# DNA binding analysis of rare variants in homeodomains reveals homeodomain specificity-determining residues

Kian Hong Kock[1,2,12], Patrick K. Kimes[3,4,12], Stephen S. Gisselbrecht [1,12], Sachi Inukai[1], Sabrina K. Phanor[1], James T. Anderson[1], Gayatri Ramakrishnan [1,5], Colin H. Lipper[6,7], Dongyuan Song [4], Jesse V. Kurland[1], Julia M. Rogers[1,8], Raehoon Jeong [1,9], Stephen C. Blacklow [2,6,7,8], Rafael A. Irizarry[10] & Martha L. Bulyk [1,2,8,9,11] ✉

Homeodomains (HDs) are the second largest class of DNA binding domains (DBDs) among eukaryotic sequence-specific transcription factors (TFs) and are the TF structural class with the largest number of disease-associated mutations in the Human Gene Mutation Database (HGMD). Despite numerous structural studies and large-scale analyses of HD DNA binding specificity, HD-DNA recognition is still not fully understood. Here, we analyze 92 human HD mutants, including disease-associated variants and variants of uncertain significance (VUS), for their effects on DNA binding activity. Many of the variants alter DNA binding affinity and/or specificity. Detailed biochemical analysis and structural modeling identifies 14 previously unknown specificity-determining positions, 5 of which do not contact DNA. The same missense substitution at analogous positions within different HDs often exhibits different effects on DNA binding activity. Variant effect prediction tools perform moderately well in distinguishing variants with altered DNA binding affinity, but poorly in identifying those with altered binding specificity. Our results highlight the need for biochemical assays of TF coding variants and prioritize dozens of variants for further investigations into their pathogenicity and the development of clinical diagnostics and precision therapies.

Understanding TF-DNA recognition is important for elucidating how TFs regulate gene transcription and how TF mutations cause disease. Members of the homeodomain (HD) class of sequence-specific TFs play important roles in regulating development, body patterning, and cellular differentiation[1,2] and contribute to a wide array of developmental disorders[3,4]. TFs of this class have been influential models of protein-DNA recognition, and many studies have analyzed HD sequence specificity[5–10], but HD-DNA recognition is still not fully understood.

Prior co-crystallographic structural studies, combined with high-throughput surveys of wildtype HDs from mouse and *Drosophila* and associated analyses of DNA binding specificities, have identified residues in the recognition helix that contact bases within the major groove of DNA and play a primary role in determining DNA binding specificity. This work has shown that HD canonical positions 47, 50, and 54 of the recognition helix (Helix 3; Fig. 1a, b) are primary specificity determining positions, as amino acid substitutions at these positions alter DNA binding site motifs[7,8,11]. Interactions of residues in the flexible N-terminal arm with bases in the DNA minor groove also contribute to specificity[12].

Numerous coding mutations within these TFs, which are found frequently within the HDs, have been associated with a wide range of

**Fig. 1 | Homeodomain variants in the human population. a** Structure of a representative homeodomain bound to DNA (PDB: 9ANT[97]). Side chains of amino acid residues reported to contact DNA are shown in stick form, while those which form the hydrophobic core of the domain are shown in space-filling view. **b** Sequence logo of the HD Pfam domain (PF00046). **c** Number of unique non-synonymous missense variants found at each HD position in gnomAD (upper panel) or ClinVar (lower panel). Source data are provided as a Source Data file[37].

congenital disorders and cancers[3,13]. Although a molecular basis for dysregulation of gene expression by some of these mutations has been determined, the effects of most HD variants on DNA binding remain unknown. Prior studies of HD-DNA binding specificities focused largely on prediction of DNA binding motif similarity based on DNA binding domain (DBD) amino acid (aa) sequence similarity[7,8,10,14–16]; however,

such approaches are limited by the diversity of protein sequences represented among the surveyed wildtype HDs and do not permit accurate prediction of the effects of single missense variants on DNA binding specificity. Furthermore, they are not designed to predict the effects of missense variants on DNA binding affinity (i.e., strength of binding). Single missense substitutions have been found to alter DNA

binding affinity, specificity, or both, including partial losses and/or gains of novel DNA recognition motifs, and have been associated with clinical phenotypes[3,17]. A number of these HD variants exhibited unanticipated effects on DNA binding, including residue substitutions at positions not known to contact DNA but that nevertheless altered DNA binding specificity[3], highlighting the need to survey a wide range of variants across the domain using a functional assay.

In this study, we analyze 92 human HD mutants, including disease-associated variants and variants of uncertain significance (VUS), for their effects on DNA binding activity using universal protein binding microarrays (PBMs) and a statistical analysis method ("upbm") that we developed in this study to detect altered DNA binding specificity or, separately, altered DNA binding affinity. Assaying numerous rare variants results in our exploring a broader sequence space of variation at various HD positions than prior studies that surveyed TF reference alleles and led to our discovery of previously unreported specificity-determining positions, including some that do not contact DNA. The upbm analysis method reveals changes in binding by HD variants to subsets of 8-mers, including DNA binding specificity changes involving 8-mers that were not among the highest affinity 8-mers bound by the corresponding reference protein. Analysis of in vivo occupancy data indicates that the lower affinity sites whose binding is affected by TF coding variants are bound in vivo. Variant effect prediction tools perform poorly in identifying variants with altered binding specificity, highlighting the need for functional assays of TF coding variants. Overall, our study provides a framework for broader investigation of coding variants of not just HDs but also other TF families.

## Results

### Survey of homeodomain missense variants' effects on DNA binding activity

To explore the range of HD missense variants across human populations, we inspected the gnomAD database[18] for variants across the exome and genome sequences of 141,456 individuals and found 4,719 unique variants across 221 HDs (Fig. 1c and Supplementary Data 1). To examine the landscape of HD coding variation identified in a clinical genetics setting, we inspected all HD variants in the ClinVar database[19]. We identified 1232 missense variants and observed that every position within the 57-aa HD Pfam domain had missense variants annotated in ClinVar as pathogenic or likely pathogenic, in addition to those annotated as a variant of uncertain significance (VUS) (Fig. 1c and Supplementary Data 1). The breadth of pathogenic variants and VUSs across the HD domain motivated us to gain a more comprehensive understanding of the potential effects of such variants, and the potential roles of their corresponding aa positions within HDs, on DNA binding activity.

In this study, we surveyed 30 HD allelic series, comprising 92 HD missense variants together with their corresponding reference allele ('wildtype') HDs, totaling 122 HD alleles, for the effects of the variants on the intrinsic DNA binding activities of these HDs (Supplementary Data 2). We focused on the ANTP/HOXL, ANTP/NKX, PRD, and SINE subfamilies since they are among the largest HD subfamilies ("Methods" section and Supplementary Fig. 1) and contain numerous disease-associated TFs. We evaluated at least 3-fold more variants than prior studies that surveyed HD missense variants[3,20]. The newly analyzed variants include known pathogenic mutations, disease-associated mutations, potentially damaging missense variants present in gnomAD, VUSs in the ClinVar database, and variants that we selected to investigate the potential roles of understudied positions within HDs on DNA binding activity. The DNA binding activity of each of the 122 HD alleles was assayed in vitro in at least duplicate on universal protein binding microarrays (PBMs) ("Methods" section; Supplementary Data 3), on which all 8-bp sequences are represented 32 times (16 times for palindromic 8-mers), resulting in a dataset of nearly 8 million unique HD-8mer evaluations.

We developed an approach (termed "upbm") to analyze the universal PBM data that combines a parametric statistical scoring of each 8-mer with criteria that evaluate the sequence similarity of significantly scoring 8-mers, to identify variants that displayed altered DNA binding affinity (i.e., strength of binding) and/or altered sequence specificity, as compared to the corresponding reference HD protein ("Methods" section; Fig. 2). Because replicate PBMs are normalized to replicates performed in parallel, the affinity estimate calculated by this approach provides a relative measure of binding to different 8-mers that can be compared between variants and their corresponding reference alleles. The parametric scoring considers the full affinity range of 8-mers, whereas the prior, non-parametric approach to identifying affinity changes considered only the top 50 scoring 8-mers[3], which correspond to the highest affinity binding sites. Lower affinity HD binding sites have been found to be important for the specificity of developmental enhancer activities in mouse[21] and *Drosophila*[22]. Furthermore, the parametric approach avoids the use of arbitrary thresholds on the difference in 8-mers' non-parametric binding scores for a reference versus variant allele in identification of altered DNA binding specificities[3,23]. In the parametric approach, for each variant HD allele each 8-mer is assigned 3 $Q$-values ("Methods" section): for significant binding (affinityQ), for significantly altered binding as compared to the corresponding reference allele (contrastQ), and for significant deviation from the trend as compared to the reference allele (specificityQ).

For validation purposes, we compared differentially bound 8-mers identified by the upbm pipeline with 8-mers previously confirmed to be bound differentially between paralogous or mutant TFs. We identified 24 published assays of DNA binding activity by coding variants tested in our dataset, representing 15 variants in 12 publications[24-35] (Supplementary Table 1). We assessed the effects shown in each publication as total loss of binding, strong loss, moderate reduction, slight reduction, or increase. We then identified, within each tested sequence, the 8mer with the highest upbm affinity estimate for the corresponding reference allele, and assessed the magnitude (contrast difference) and significance (contrastQ) of the change in binding to that 8mer caused by the coding variant. The 19/24 published experiments with moderate, strong, or total loss of binding all showed highly significant ($P < 10^{-4}$) negative contrast differences. By comparison, the 5/24 experiments with slight reduction or increase of binding did not show significant contrastQ values. Analysis by upbm of PBM data for the paralogous *S. cerevisiae* zinc finger TFs Msn2, Msn4, Com2, Usv1, and Rgm1[36] successfully identified alternate DNA binding sequences bound preferentially by Com2 and Usv1 as previously confirmed by electrophoretic mobility shift assays (Supplementary Fig. 2a), recapitulated that Msn4 has the same binding specificity as Msn2 (Supplementary Fig. 2b–e), and confirmed 8-mers affected by mutations in the Msn2, Com2, or Usv1 DBDs (Supplementary Fig. 2f–i). In addition, results from upbm analysis of PBM data for wild type and two mutants of HOXD13 – Q325R and Q325K (affecting HD canonical position 50, hereafter abbreviated "HD50") – were concordant with HOXD13 ChIP-Seq data[17] on the genomic occupancies of these HOXD13 alleles (Supplementary Fig. 2j–m).

We set $Q < 10^{-6}$ thresholds on each of these metrics for identifying individual 8-mers with altered DNA binding affinity and/or specificity. We then set 5% false discovery rate (FDR) thresholds for calling variant HDs with significantly altered DNA binding affinity or specificity, based on the number of altered 8-mers meeting our $Q$-value and sequence similarity criteria ("Methods" section, Supplementary Fig. 2n,o) in negative control comparisons of reference-versus-reference replicates. Using the upbm approach, we analyzed PBM data for 66 variants assayed in this study that had not been assayed in prior studies, and re-analyzed data for 26 variants generated in a prior study[3] (Supplementary Data 4[37]). The majority of re-analyzed datasets produced identical calls for affinity and/or specificity alteration with either criteria, but 6 variants are called affinity-reducing and 5 are called

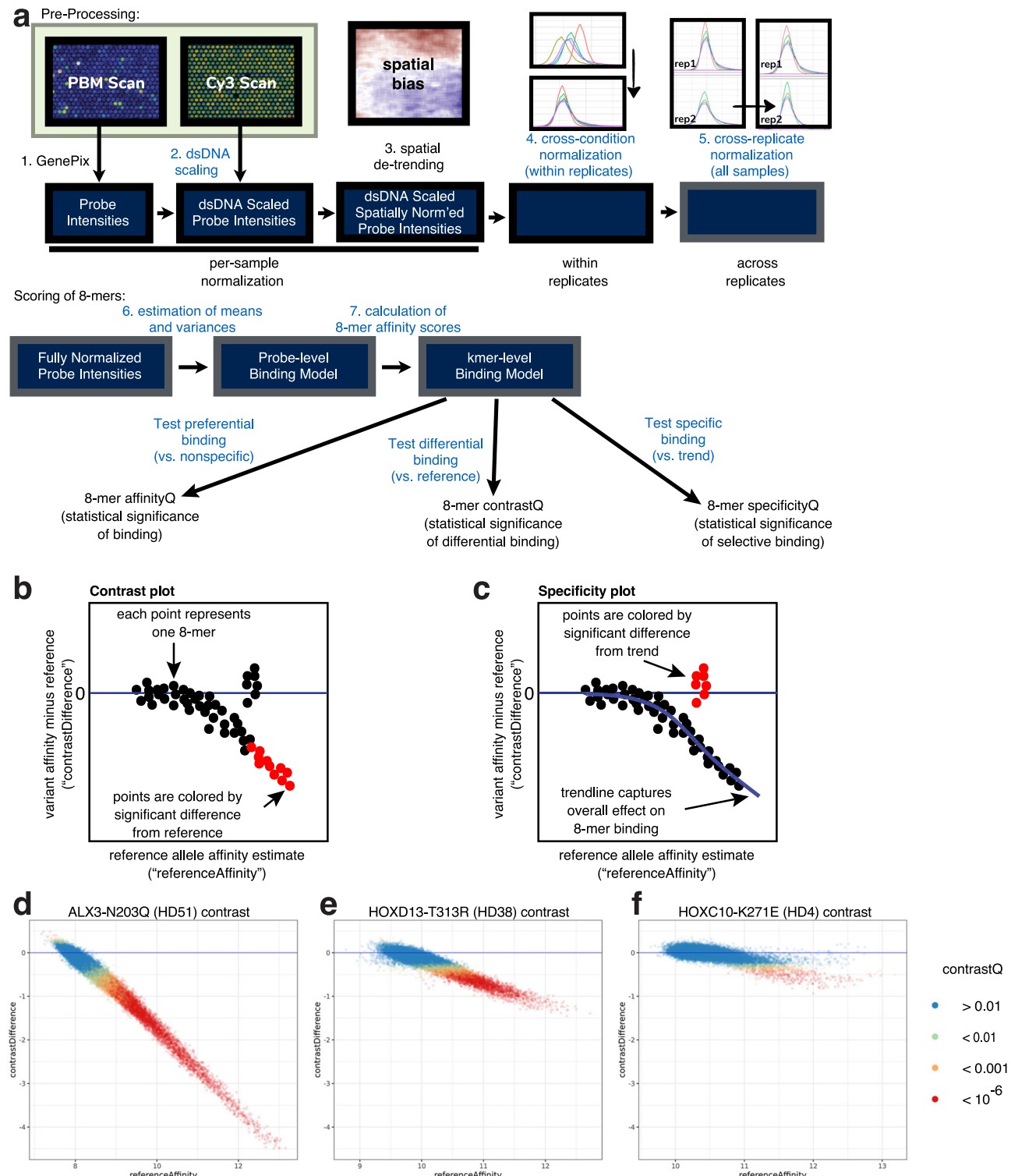

**Fig. 2 | The upbm pipeline for analysis of variant effects. (a)** Schema of upbm analysis. Steps indicated in black are part of the previously published workflow[82,83]; steps indicated in blue were developed in this study. **(b,c)** Effects on binding calculated with the upbm package are visualized by plotting contrast difference − the affinity score for the variant minus the reference allele's affinity score − versus the reference affinity estimate. **b** In contrast plots, points are colored by "contrastQ," the statistical significance of differential binding vs. the reference. **c** In specificity plots, a trendline is fitted to the data (B-spline fit; see "Methods" section) and points are colored by the significance of divergence from the trendline, reflecting specific effects on binding to subsets of 8-mers. **d−f** Representative examples of (**d**) a variant which causes severe, uniform loss of binding, (**e**) one causing milder uniform loss of binding, and (**f**) one which differentially affects binding to a subset of 8mers. Source data are provided as a Source Data file[37].

specificity-altering (Supplementary Data 2); these variants' affinity-reducing and specificity-altering properties had not been identified before. These typically showed alterations preferentially affecting 8-mers in the lower portion of the reference allele's affinity range, which our prior methods tended to de-emphasize. In addition, one variant previously called affinity-reducing and 3 variants previously called specificity-altering do not meet our current criteria and are reclassified as "no change" (Supplementary Data 2). These were

associated with smaller effect sizes and may reflect a conservative threshold based on our understanding of the noise distribution in PBM data learned from the negative control replicates (see "Methods" section).

## Performance of variant effect prediction tools in discriminating variants with altered DNA binding activity

Using our integrated set of criteria to analyze the HD variants' PBM data, we identified 51 variants with altered DNA binding affinity and 28 with altered specificity (Fig. 3a; Supplementary Data 2; Supplementary Fig. 3). Variants with altered DNA binding activity were enriched for pathogenicity in the ClinVar and ADDRESS databases[19,38] ($P = 0.0203$, two-tailed Fisher's exact test). We investigated how well the altered DNA binding activities of HDs could be predicted by variant effect prediction tools. We evaluated 42 different variant interpretation tools that utilize a range of prediction algorithms. We reasoned that altered DNA binding affinity, which in nearly all cases corresponded to decreased affinity, represents a different type of damaging effect, often involving different mechanisms, than altered DNA binding specificity. Therefore, we separately analyzed the ability of the tools to distinguish 46 variants with altered affinity and 23 variants with altered specificity (17 variants were in both sets), from those that did not show such altered activity, respectively. Multiple tools achieved moderately good accuracy in distinguishing variants with altered DNA binding affinity, among which MutationAssessor[39], MetaLR[40], ClinPred[41], and SIFT[42] were the top performers (area under receiver operating characteristic curve (AUROC) = 0.86, 0.81, 0.80, and 0.79, respectively), while some popular tools did not perform well (e.g., AlphaMissense[43], 0.69, CADD[44], 0.63); numerous methods, particularly those based primarily or solely on evolutionary sequence conservation (e.g., MutationTaster[45], PhastCons[46]), performed poorly (AUROC < 0.5) (Fig. 3b; Supplementary Fig. 4a, b; and Supplementary Data 5). In contrast, none of the 42 tools was able to accurately distinguish variants that altered DNA binding specificity, with the highest AUROC among the tools being 0.66 (MetaSVM) (Fig. 3c; Supplementary Fig. 4c, d; and Supplementary Data 5). Overall, considering all variants with either type of altered DNA binding activity, MutationAssessor was the top performing algorithm (AUROC 0.86; Fig. 3d and Supplementary Fig. 4e, f). We note that while MutationAssessor, MetaLR, ClinPred, and SIFT performed well in distinguishing variants with altered affinity, they performed poorly (AUROC = 0.58, 0.48, 0.54, and 0.55, respectively) in distinguishing those with altered specificity.

## Identification of disease-associated rare homeodomain variants with altered DNA binding affinity

Twenty-two of 29 assayed known pathogenic or disease-associated mutations exhibited reduced DNA binding affinities (Table 1). For example, the L168V mutation in ALX3 (HD16), which causes a spectrum of frontonasal deformities[47], was identified in our analysis as showing reduced binding affinity (Fig. 4a). Three of the 8 assayed VUSs in the ClinVar database exhibited reduced affinity. For example, HOXA4 R217L (HD3), which has been identified with unclear clinical significance in one individual, shows reduced binding affinity and modestly altered specificity (Fig. 4b and Supplementary Fig. 4ax). Finally, 16 of 32 missense variants that were present only in gnomAD also showed reduced affinity. For example, the rare variant NKX2-6 R150C (HD19; MAF 3.2 ×10[−5]), which does not occur at a DNA-contacting position, showed diminished affinity (Fig. 4c); other substitutions in NKX2-6 have been found associated with septation defects in the heart[48,49]. For comparison, the common variant SIX6 H141N (HD14; MAF 0.52), which is classified as benign in the ClinVar database, shows no significant change in DNA binding activity (Fig. 4d). A single rare variant, NKX2-4 R246Q (HD58; MAF 4.25 ×10[−6]) showed an increase in binding affinity (Fig. 4e).

Altogether, 51 variants, spanning 25 aa positions throughout the HD domain, altered DNA binding affinity in our survey; all but one of

these decreased binding affinity (Supplementary Data 2 and Supplementary Fig. 3). Forty-seven of these are either at likely DNA-contacting positions or in the hydrophobic core, where mutations may destabilize the protein. Our results support prior low-throughput studies that reported decreased DNA binding affinity by the variants ARX T333N (HD6) and P353R (HD26) at DNA-contacting positions, which cause developmental brain malformations[24]. The remaining 4 of the affinity-altering variants, at 4 different positions, suggest other residues that may be important for protein folding or stability. Three of these variants, ALX4 Q225E (HD12, a variant that causes frontonasal dysplasia[50]), NKX2-6 R150C (HD19), and PITX2 R108H (HD24), are charge-altering substitutions in surface-exposed residues; such residues have been suggested to impact stability of small globular domains[51]. The fourth, PITX2 T114P (HD30), is a proline substitution in helix 2; as proline geometrically restricts the protein backbone and lacks a backbone hydrogen bond donor for an alpha-helical conformation, this variant would be expected to disrupt the folding or stability of the domain. In addition, our results indicate that multiple variants reduce binding to lower affinity 8-mers much more than to the most preferred sites, such as HESX1 R109Q (HD2), which causes combined pituitary hormone deficiency[34] (Fig. 4f; compare highest affinity [blue arrowhead] to more moderate affinity 8-mers [black arrowhead]).

## Disease-associated homeodomain variants exhibit a spectrum of different types of altered homeodomain DNA binding specificities

Nine of 29 assayed known pathogenic or disease-associated mutations exhibited a variety of different types of altered DNA binding specificity (Table 1; Supplementary Fig. 3). For example, whereas HOXD13 Q325R (HD50), which causes syndactyly[52], resulted in a change of specificity with a loss of binding to some recognition sequences and a gain of recognition of a novel motif[3], ALX4 Q225E (HD12) showed overall reduced DNA binding affinity with preferential retention of the highest affinity 8-mers and a specific set of lower affinity sites (Fig. 5a). In addition, we identified 9 variants in 8 known disease-associated genes that showed altered DNA binding activity (Table 1 and Supplementary Fig. 3). For example, various mutations in MSX2 have been found to cause craniosynostosis and foramina[53]. We found the rare MSX2 variant R199I (HD58; MAF 3.98 × 10[−6]) to cause a reduction in DNA binding affinity; the R199I variant also altered binding specificity by preferentially reducing binding to lower affinity 8-mers containing the alternate motif CAATCA (Fig. 5b). Finally, we found variants in PBX4, which is expressed in a range of tissues including lymphocytes, lymph nodes, and the thyroid[54,55]. In genome-wide association studies (GWAS), *PBX4* has been found associated with various disease-associated traits, including lymphocyte counts[56], triglycerides, and LDL cholesterol[57], but no pathogenic protein-coding variants have been identified. We found the PBX4 rare variant R215Q (HD6; MAF 3.33 x 10[−4]) to exhibit a severe loss of DNA binding affinity.

The upbm parametric approach allowed for more robust evaluation of changes in binding by a variant HD to subsets of 8-mers, including DNA binding specificity changes involving 8-mers that were not among the highest affinity 8-mers bound by the corresponding reference protein (Supplementary Fig. 3). For example, PROP1 R112Q (HD44) altered DNA binding specificity with a gain of moderate affinity sites (Fig. 5c); our upbm analysis method was able to detect this altered specificity, which was missed in our prior study that was focused on the highest affinity binding sequences[3]. As another example, we identified a gain in specificity by the SIX6 T165A (HD38) variant for a recognition motif that comprises 8-mers bound with lower affinity by the reference SIX6 protein. This variant was found in a patient with microphthalmia and cataracts[58] and was initially called a putatively pathogenic variant in the Online Mendelian Inheritance in Man (OMIM) database[59], but in both OMIM and ClinVar had been reannotated as "Reclassified –

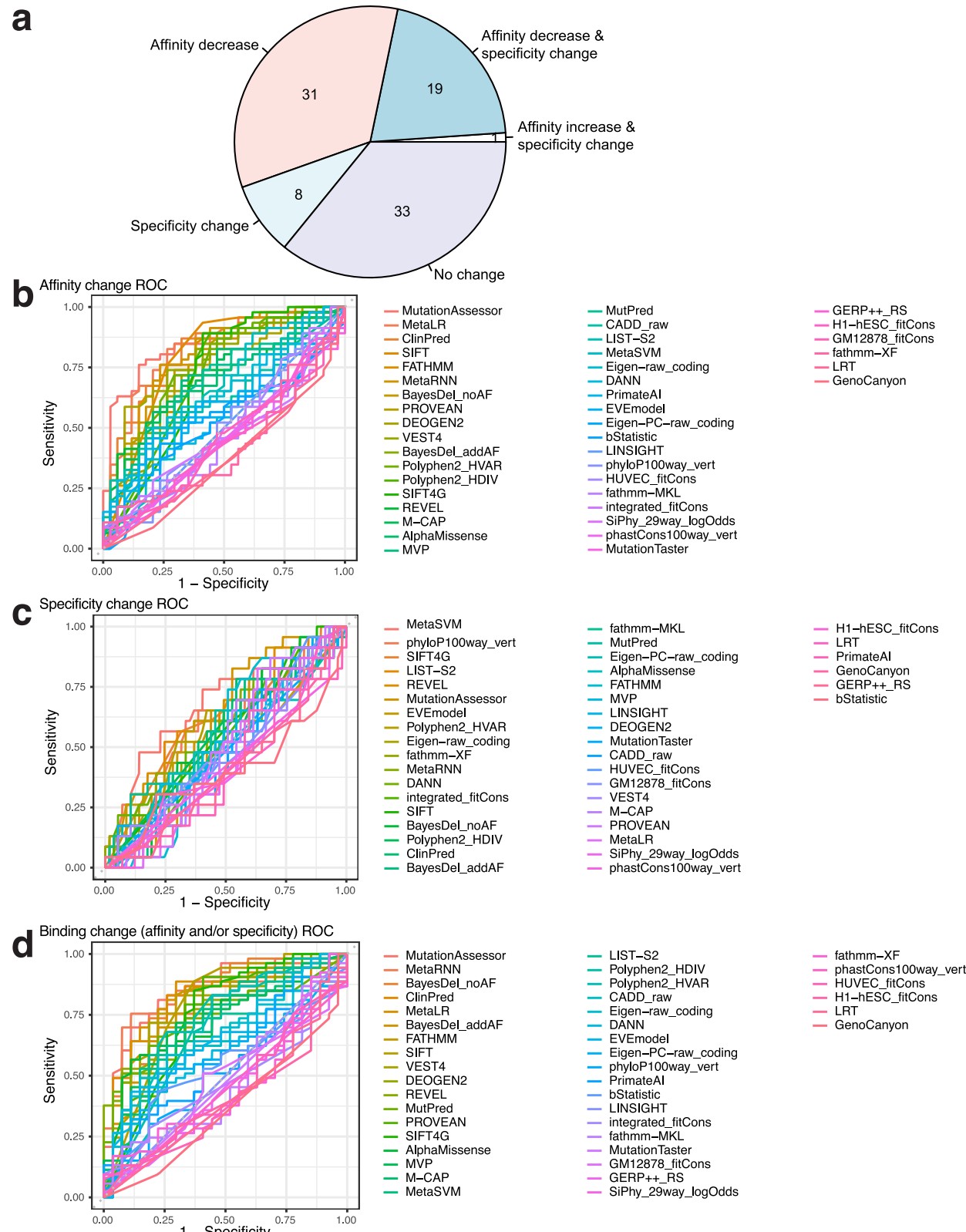

**Fig. 3 | Homeodomain variants with altered DNA binding affinity or specificity.**
**a** Pie chart showing the number of tested variants that altered DNA binding affinity, specificity, both, or neither. **b**–**d** Receiver operating characteristic (ROC) curves for 42 variant effect prediction tools assessing their ability to discriminate HD variants with altered (**b**) affinity, (**c**) specificity, (**d**) or either. Precision-recall curves and area under the curve statistics for all prediction tools are shown in Supplementary Fig. 4. Source data are provided as a Source Data file[37].

**Table 1 | Altered DNA binding activities not previously identified for naturally occurring homeodomain variants in disease-associated genes**

| Gene | Variant | Binding change | Known causal variant | Disease association | Reference |
|---|---|---|---|---|---|
| ALX1 | L144R | Affinity change | No | Frontonasal dysplasia 3 | PMID: 2045171 |
| ALX3 | L168V | Affinity change | Yes | Frontonasal dysplasia 1 | PMID: 19409524, 17963218 |
| ALX3 | N203S | Affinity change | Yes | Frontonasal dysplasia 1 | PMID: 19409524, 17963218 |
| ALX4 | Q225E | Affinity and specificity change | Yes | Frontonasal dysplasia with alopecia and genital anomaly | PMID: 22140057 |
| ARX | T333N | Affinity change | Yes | Agenesis of the corpus callosum, with abnormal genitalia | PMID: 14722918 |
| HESX1 | R109Q | Affinity change | Yes | Combined pituitary hormone deficiency | PMID: 28332357, 26781211 |
| HESX1 | F115L | Affinity change | No | Growth hormone deficiency with pituitary anomalies; Septooptic dysplasia; Pituitary hormone deficiency, combined, 5 | PMID: 9620767, 16940453, 11136712 |
| HOXA2 | Y150S | Affinity change | No | Microtia with or without hearing impairment | PMID: 23775976 |
| HOXD13 | R306W | Affinity and specificity change | Yes | Synpolydactyly type 1 | PMID: 12414828 |
| HOXD13 | T313R | Affinity change | Yes | Synpolydactyly | PMID: 26581570 |
| HOXD13 | I322L | Specificity change | Yes | Brachydactyly types D and E | PMID: 16314414, 12649808, 12620993 |
| MSX2 | R199I | Affinity and specificity change | No | Craniosynostosis 2; Parietal foramina 1; Parietal foramina with cleidocranial dysplasia | PMID: 8357019, 10742103, 10767351, 14571277 |
| NKX2-5 | K183E | Affinity and specificity change | Yes | Atrioventricular septal defect, somatic | PMID: 15917268 |
| NKX2-5 | R190C | Affinity and specificity change | Yes | Atrial septal defect 7 | PMID: 15364612, 15810002, 20456451 |
| NKX2-6 | F151L | Affinity change | Yes | Persistent truncus arteriosus | PMID: 15649947 |
| NKX2-6 | R150C | Affinity change | No | Persistent truncus arteriosus | PMID: 15649947 |
| PAX4 | F177V | Affinity and specificity change | No | Diabetes mellitus, type 2; Maturity-onset diabetes of the young, type IX | PMID: 11723072, 17426099 |
| PAX4 | R227W | Affinity change | No | Diabetes mellitus, type 2; Maturity-onset diabetes of the young, type IX | PMID: 11723072, 17426099 |
| PBX4 | R215Q | Affinity and specificity change | No | Blood lipid levels; blood cell counts | PMID: 34610981, 30275531, 32888493, 27863252 |
| PITX2 | R108H | Affinity and specificity change | Yes | Ring dermoid of cornea | PMID: 15591271 |
| PITX2 | T14P | Affinity change | Yes | Axenfeld-Rieger syndrome type 1 | PMID: 8944018, 9685346, 18723525 |
| PROP1 | F88S | Affinity change | Yes | Pituitary hormone deficiency, combined, 2 | PMID: 10946881, 20301521 |
| PROP1 | R99Q | Affinity change | Yes | Pituitary hormone deficiency, combined, 2 | PMID: 12519826, 18157385 |
| PROP1 | R112Q | Specificity change | No | Congenital hypogonadotropic hypogonadism | PMID: 32982993 |
| PRRX1 | F113S | Affinity change | Yes | Agnathia-otocephaly complex | PMID: 12244557, 21294718 |
| SHOX | R173C | Affinity change | Yes | Leri-Weill dyschondrosteosis | PMID: 12116253, 15931687, 28973083 |
| SIX6 | T165A | Specificity change | No | Optic disc anomalies with retinal and/or macular dystrophy | PMID: 15266624, 23167593, 24702266 |

For each variant, we report the observed DNA-binding change (affinity, specificity, or both); whether the variant has itself been reported to be pathogenic; the disease or diseases associated with the variant or gene, and selected references.

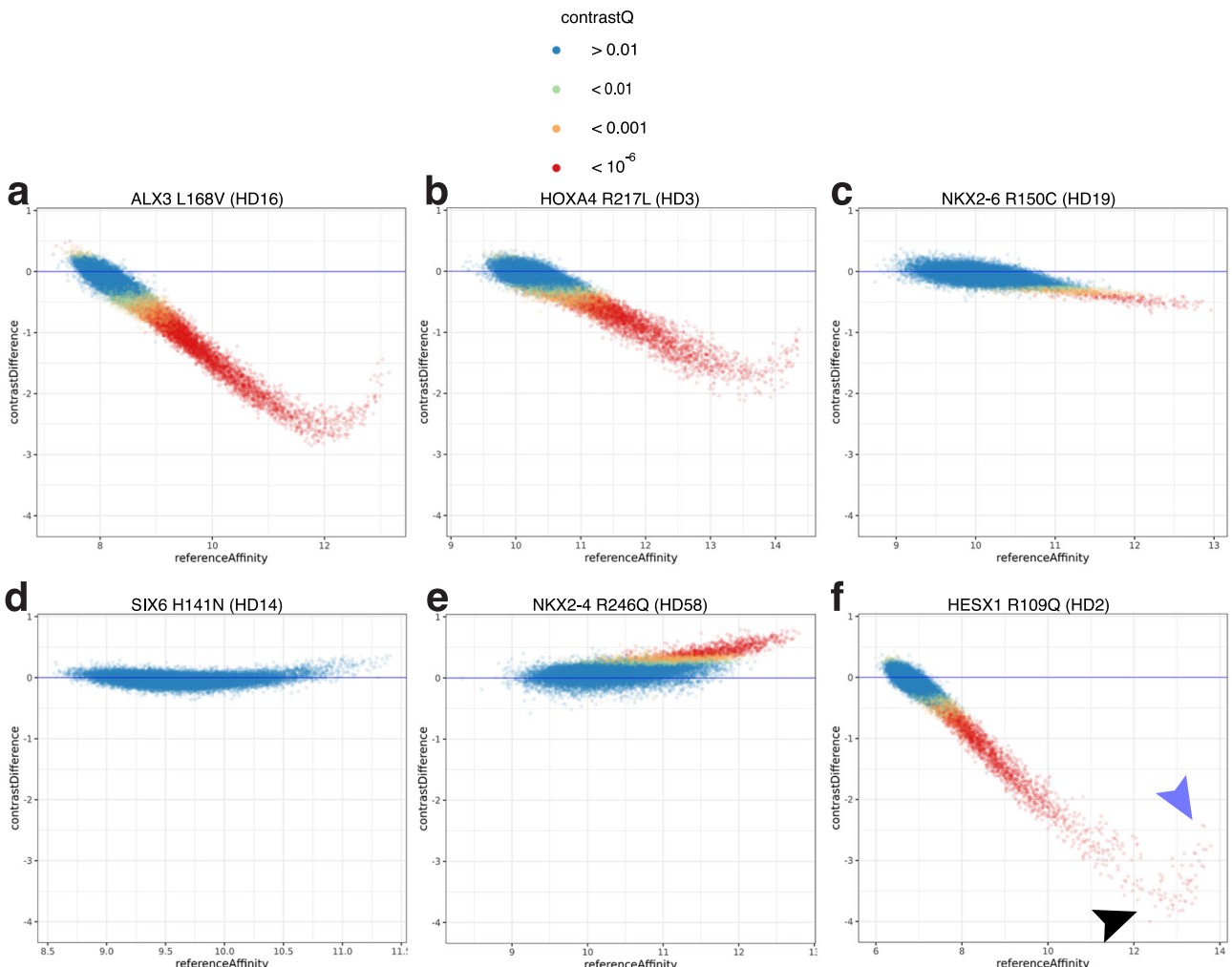

**Fig. 4 | Known or putatively pathogenic variants show a range of DNA binding affinity alterations.** Contrast plots as described in Fig. 2b. **a** ALX3 L168V shows reduced affinity across all significantly bound 8-mers. **b** HOXA4 R217L shows both affinity and specificity changes. **c** NKX2-6 R150C shows diminished binding affinity. **d** SIX6 H141N shows no significant change in binding. **e** NKX2-4 R246Q shows increased binding affinity. **f** HESX1 R109Q has strongly reduced binding affinity; 8-mers bound with the highest affinity by the reference allele (blue arrowhead) are less affected than moderate affinity 8-mers (black arrowhead). Source data are provided as a Source Data file[37].

Variant of Uncertain Significance". While SIX6 T165A and the reference SIX6 protein both bound with highest affinity to a common set of 8-mers corresponding to a TGATAC motif, the SIX6 T165A variant also gained recognition of a TGA**C**AC motif (Fig. 5d). We identified a similar gain in recognition by the synthetically designed SIX6 T165G mutant (Fig. 5e). We analyzed ChIP-seq data for murine SIX6[60], which is identical to human SIX6 in protein sequence throughout the region analyzed by PBMs, ectopically expressed in NIH-3T3 cells, and found TGACAC to be significantly enriched (vs. a matched background created with the GENRE package[61]; see "Methods" section) in the top 1000 ChIP-seq peaks ($P = 2.11 \times 10^{-6}$, one-tailed Fisher's exact test; Fig. 5f). These results suggest that altered affinity of SIX6 T165A for these sites may be relevant to its function in vivo. Variants at other aa positions in other HDs also showed changes in binding to subsets of lower affinity 8-mers; some of the variants (e.g., VENTX R121Q [HD31] and VENTX Y115F [HD25], which have not been associated with disease or trait variation), while resulting in an overall decrease in DNA binding affinity, showed a preferential loss of binding affinity to a subset of lower affinity 8-mers (Fig. 5g,h). These two VENTX variants both preferentially reduce binding to 8-mers containing a CGATTA motif. An experimentally determined structure is not available for VENTX, but the analogous (identical) residues in NKX2–5, a homologous member

of the ANTP/NKL HD subfamily ("Methods" section and Supplementary Table 2), appear to make adjacent contacts to the DNA backbone (Fig. 5i)[62]. Similarly, CRX R41L (HD3) caused a preferential reduction in binding to the lower affinity TAAGCC motif, compared to the optimal TAATCC (Fig. 5j). The TAAGCC motif is highly enriched (vs. GENRE background) in the top 1,000 CRX ChIP-seq peaks[63] ($P < 2.2 \times 10^{-16}$, one-tailed Fisher's exact test; Fig. 5k), suggesting that CRX mutants that impair recognition of this motif could be relevant to its function in vivo.

**Missense variant analysis reveals 14 previously unknown homeodomain DNA binding specificity-determining residues**

Altogether, the 92 variants we evaluated in this study covered 35 HD aa positions, including many for which potential contributions to DNA binding specificity were unanticipated or had not been investigated in depth. Our analysis revealed 14 HD aa positions that to our knowledge were not described previously in the literature as specificity-determining positions but at which we found missense variants that altered DNA-binding specificity (Fig. 6a, b and Supplementary Data 2). These specificity-determining positions correspond to canonical HD aa positions 4, 12, 16, 24, 25, 26, 31, 38, 40, 44, 46, 48, 53, and 58 (note: canonical HD positions correspond to Pfam position plus 1). Position 4

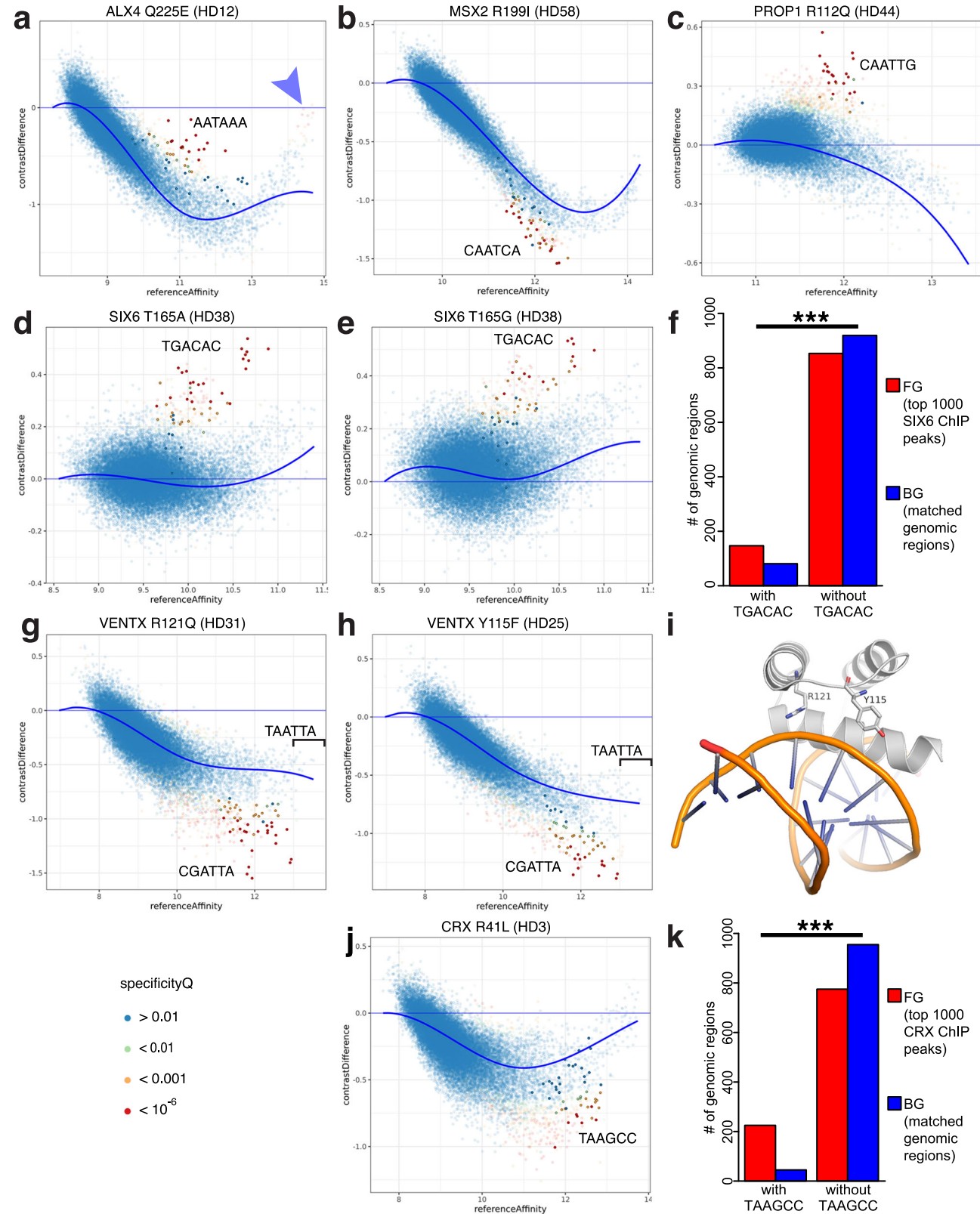

is in the N-terminal arm, positions 12 and 16 are in Helix 1, positions 24–26 are in the linker between Helix 1 and Helix 2, position 31 is in Helix 2, Positions 38 and 40 are in the linker between Helix 2 and Helix 3, and positions 44, 46, 48, 53, and 58 are in Helix 3 (the 'recognition helix').

To gain insights into the mechanisms by which these HD positions and associated variants influenced DNA binding specificity, we performed homology modeling of the DNA-bound HD ("Methods" section and Supplementary Table 2). Although position 58 has been reported to interact with the major groove and the DNA phosphate backbone in HD-DNA structures available in PDB[3,10], it was found in prior bioinformatic studies to either not be associated[15], or show minimal association, with HD DNA binding specificity[10]. Of the 5 different missense substitutions that we tested at this position within 5

**Fig. 5 | Variants with altered DNA binding specificity due to preferential effects on binding to lower affinity binding sites.** Specificity plots as described in Fig. 2c. **a** ALX4 Q225E reduces binding to moderate affinity 8-mers, but selectively spares those containing AATAAA (highlighted). **b** MSX2 R199I preferentially reduces binding to 8-mers containing CAATCA. **c** PROP1 R112Q causes a gain of binding to 8-mers containing CAATTG. **d**, **e** SIX6 T165A and T165G cause similar gains of binding to 8-mers containing TGACAC; (**f**) TGACAC is significantly enriched ($P = 2.11 \times 10^{-6}$) in top 1000 ChIP-seq peaks for murine SIX6 ectopically expressed in NIH-3T3 cells (FG) vs. matched genomic background (BG). **g**, **h** VENTX R121Q and Y115F cause similar selective loss of binding to 8-mers containing CGATTA; (**i**) these two HD residues make adjacent backbone contacts in the structural model built from a published NKX2-5:DNA co-crystal structure (PDB: 3RKQ[98]). **j** CRX R41L preferentially reduces binding to 8-mers containing TAAGCC; (**k**) this 6-mer is significantly enriched ($P < 2.2 \times 10^{-16}$) among top 1000 reference CRX ChIP-seq peaks (FG) vs. matched genomic background (BG). ***$P < 10^{-3}$, one-sided Fisher's exact test. Source data are provided as a Source Data file[37].

HDs from 2 HD subfamilies, only the rare variants MSX2 R199I and NKX2-4 R246Q altered binding specificity, suggesting that this position may exert a context-specific effect that depends on the particular HD and/or amino acid substitution; 2 of the other 3 variants (HESX1 R165K and VAX1 K157R) corresponded to more physicochemically conservative substitutions, and the remaining one, PAX4 R227W, resulted in reduced DNA binding affinity, as did MSX2 R199I. Our homology modeling results suggest that the reference amino acid contacts a DNA base in MSX2 and the phosphate backbone in NKX2-4 and PAX4, and that these three variants result in a change (NKX2-4 R246Q) or loss of contacts (MSX2 R199I, PAX4 R227W), consistent with our observed results. At position 44, which has been reported to be a backbone-contacting position[3,10], we found that the rare variant PROP1 R112Q altered DNA binding specificity with a gain of moderate affinity sites (Fig. 5c). The possibility that recognition of particular low-affinity binding sites depends on specific backbone contacts suggests the involvement of indirect readout of the underlying DNA sequence through its effects on DNA shape[64]. Similarly, at the backbone-contacting position 4, of the 4 different missense substitutions that we tested within 4 HDs from 2 HD subfamilies, only the rare HOXC10 variant K271E, which changes a basic amino acid to an acidic one, altered binding specificity, suggesting that this position may exert a context-specific effect that depends on the particular HD and/or amino acid substitution; 2 of the other 3 variants corresponded to physico-chemically conservative substitutions, while the remaining one, HESX1 P111T, resulted in a moderate loss of DNA binding affinity.

Most of the linker between Helix 1 and Helix 2 is not located near the DNA in available HD-DNA structures, although a few positions are near the phosphate backbone. Position 25 and to a lesser extent position 26 contact the DNA phosphate backbone[3,10]. At position 25, the variant VENTX Y115F altered DNA binding specificity, likely through loss of a backbone contact resulting in the observed selective loss of binding to a subset of moderate affinity 8-mers. At position 26, we assayed P353L and P353R mutations in ARX. While P353L strongly reduced affinity and altered specificity, P353R only moderately reduced affinity; these results suggest that the differences in these mutations' effects on DNA binding activity may underlie the differences in their disease phenotypes, with P353R associated with X-linked lissencephaly and P353L associated with mental retardation, and correlate with the neurological phenotypes in the corresponding knock-in mouse models[65].

Position 31, a DNA phosphate backbone-contacting position in Helix 2[3,10], was found in prior bioinformatic studies to either not be associated[15], or show minimal association[10], with HD DNA binding specificity[10]. Two of 7 mutants that we tested at this position in 5 HDs – VENTX R121Q and HOXD13 R306W (a known disease mutation found in synpolydactyly patients[66]) – altered both DNA binding specificity and affinity. The observed selective loss of binding to a subset of moderate affinity 8-mers is likely due to loss of a phosphate backbone contact made by the mutated Arg residue for both of these mutants.

Since Helix 1 is located distal to the DNA in available HD-DNA structures, our finding that ALX4 Q225E (HD12) resulted in altered DNA binding specificity was particularly surprising. Similarly, the C-terminal portion of Helix 2 does not contact the DNA, and so likewise we were surprised to find that at position 38 SIX6 T165A, which has been found in a human microphthalmia patient[58], altered specificity. Residues in

the ETS TF ELK1 distal to DNA in the ELK1-DNA complex have been found to contribute to ELK1 DNA-binding specificity[67]. To our knowledge, however, such effects have not been reported previously for HDs. Our homology modeling results suggest that SIX6 Thr-165 at position 38 contacts Thr-139 at position 12, Leu-143 at position 16, and Leu-167 at position 40 (Supplementary Fig. 5); intriguingly, these residues are within, or adjacent to, the hydrophobic core, and thus are not ones that would have been expected a priori to contribute to DNA binding specificity. Our results suggest that perturbations to this network of amino acid interactions might alter the positioning of the recognition helix in the DNA major groove. To test this model, we created a set of synthetic mutants designed to represent potentially informative aa substitutions at these positions. SIX6 T165G and T165P also exhibited specificity changes (Supplementary Fig. 5). Notably, both SIX6 L167M and L167I at position 40 and L143V at position 16 showed a selective loss of binding to subsets of 8-mers (with L143V also showing overall reduced binding affinity), supporting the importance of these positions for SIX6 DNA binding specificity. Intriguingly, 8-mers containing the 7-mer GGTGTCA showed a gain in binding by SIX6 T165A and T165G (Fig. 5c, d), but a selective loss in binding affinity by SIX6 L167 and L143 mutants. Future studies are needed to determine the mechanisms of such altered binding activities.

Expanding on prior observations of different gains and losses of recognition by HOXD13 Q325R versus Q325K at HD position 50 in the recognition helix[3], we observed additional cases of physicochemically similar substitutions at the same position in the same HD showing different effects on DNA binding activity. For example, substitution of Leu-143 in SIX6 to either Val or Ile, at HD position 16 in Helix 1, both reduced DNA binding affinity, but only L143V altered DNA binding specificity (Supplementary Data 2). These results highlight the need to assay each variant, rather than inferring functional consequences on protein activity from similar substitutions.

Variation in which amino acids contact DNA in available HD-DNA co-crystal structures exists across HDs[3]. A computational analysis of mouse HD PBM data inferred the importance of different HD aa positions for DNA binding specificity, with differences among Hox, Nkx, Dlx, Lhx, Obox, or TALE HDs[16]. Therefore, we examined whether the same missense substitution at analogous HD aa positions resulted in similar effects on DNA binding activity across HD family members. Indeed, we observed multiple instances of differences in the effects of a given missense substitution on the DNA binding activities of different HD family members. For example, Arg-to-Gln substitution at HD canonical position 31, a DNA phosphate backbone-contacting position that we found to be a DNA specificity determining position (Fig. 5g), resulted in changes in HD DNA binding specificity and affinity in VENTX, a member of the ANTP/NKL subfamily, a change in only affinity in PROP1, a PRD subfamily member and no change in specificity or affinity in the PRD subfamily member PAX4 (Fig. 6c–e). Similarly, Thr-to-Ala substitution at HD canonical position 38, another DNA specificity determining position that we identified (see above), resulted in no change in DNA binding activity in HOXD13, an ANTP/HOXL subfamily member, but altered specificity in SIX6, a SINE subfamily member (Supplementary Fig. 6a,b). Strikingly, we observed several cases of the same missense substitution at the same position in the HD causing different effects in different members of the same HD subfamily

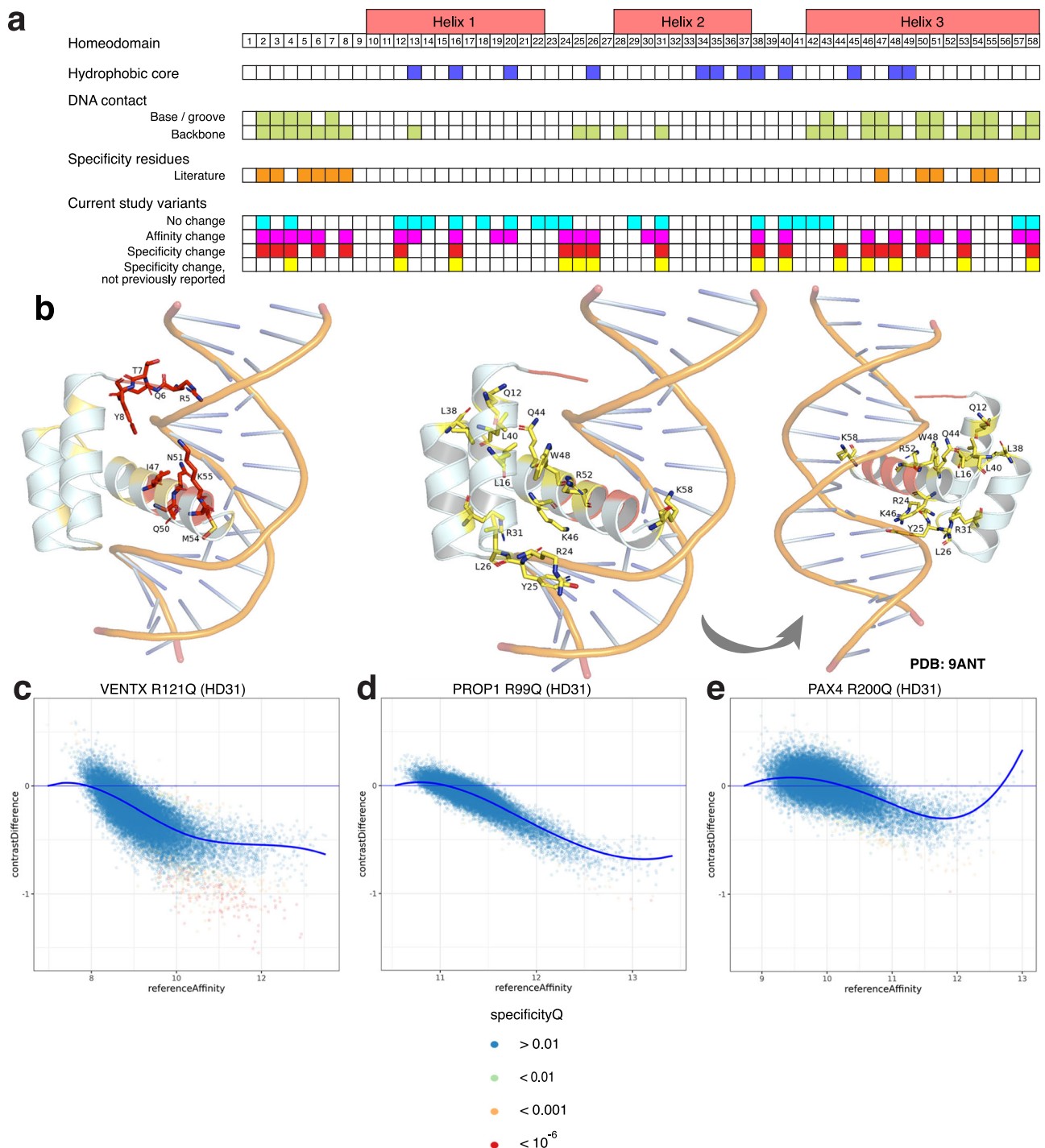

**Fig. 6 | Overview and context-dependence of HD-DNA binding specificity-determining positions. a** Summary of locations of variants discovered in this study to alter DNA binding activity, with comparison to HD structural features, DNA contacts, and previously described specificity-determining positions. **b** Specificity-determining positions that were either previously known (red) or discovered in this study (yellow) are mapped onto a structure of the Antp HD bound to DNA (PDB: 9ANT[97]). Side chains of previously described specificity-determining positions (left). Side chains of positions discovered in this study to influence specificity, shown in two views (center, right). **c–e** Arginine to glutamine mutations at canonical HD position 31 have different effects in different HD contexts, altering affinity and specificity in VENTX, altering affinity only in PROP1, and causing no significant change in PAX4 DNA binding. Source data are provided as a Source Data file[37].

(Supplementary Data 2). For example, Arg-to-Trp substitution at canonical position 31 resulted in strongly reduced affinity in HOXC9, an ANTP/HOXL HD, but mildly reduced affinity and altered specificity in HOXD13, another ANTP/HOXL HD (Supplementary Fig. 6c, d). As another example, Leu-to-Gln substitution at position 16 conferred different effects on DNA binding affinity in different members of the PRD subfamily; it showed no effect in PITX2 but resulted in strongly

decreased affinity in ARX and ALX3 (Supplementary Fig. 6e–g), suggesting it destabilizes the hydrophobic core in specific subfamily members.

## Discussion

In this study, we surveyed variants spanning a wide range of HD aa positions for their potential effects on DNA binding affinity or

specificity. While prior large-scale experimental surveys of HD-DNA recognition have focused on determining HD DNA binding preferences and inferring aa residues associated with DNA binding specificity[7,8], trait-associated variants that impact DNA binding activity can have altered DNA binding specificity or affinity, either of which can result in dysregulation of gene expression programs. Identifying residues that modulate DNA binding specificity and/or affinity will allow for a better understanding of how HD-DNA recognition is achieved and will enable improved interpretation and prediction of the effects of clinical variants. Here, we identified 46 naturally occurring (non-synthetic) HD variants with altered DNA binding activity, including a range of types of loss and/or gain of DNA binding.

The comprehensive survey of binding to all possible 8-bp sequences combined with our upbm data analysis approach allowed us to identify DNA binding specificity changes that affected only subsets of binding sites which may have been missed in prior studies that focused on effects on larger numbers of sites and/or higher affinity sites[8,10,68]; it revealed that some pathogenic variants or VUSs (e.g., SIX6 T165A) specifically altered binding to lower affinity DNA sequences. Although studies often focus on higher affinity sites, lower affinity sites can also play important roles in gene regulation; lower affinity sites are bound in vivo by various TFs[7,22], regulate gene expression in a spatiotemporal-specific manner[21,22,69], and provide robust enhancer activity[22]. Lower affinity sites and sites corresponding to previously uncharacterized motifs bound by gain-of-recognition mutants are occupied in cells according to ChIP-Seq data (Fig. 5f,k). Thus, pathogenic effects may be due to partial loss-of-function and/or gain-of-recognition in DNA binding activity. Lower affinity binding sites are captured more accurately by k-mer (here, 8-mer) DNA binding data than by a PWM that represents a TF's overall DNA binding preference. Future studies are needed to determine the impact on gene regulatory networks of clinically relevant variants found in this study to have altered DNA binding activity.

The ability to predict the DNA-binding consequences of missense variants in TFs can aid in exome and genome interpretation. We identified thousands of HD missense variants present in the human population, including hundreds of ClinVar variants of uncertain significance (Supplementary Data 1), highlighting the need to prioritize biochemically damaging variants[70] for further investigation into mechanisms underlying human disease. In particular, variants found only in large-scale sequencing studies of populations, including ostensibly healthy individuals, are of unknown relevance for human health. We suggest that such TF variants that are shown to be damaging[70] to their DNA binding activity are higher priority candidates for further study for their potential contribution to disease or other traits. The missense mutations that we found to damage DNA binding activity in the assayed HDs may also be damaging at the same positions in other HDs; however, experimental testing is needed to verify their effects. In addition, physicochemically similar substitutions at the same position in the same HD showed different effects on DNA binding activity.

While some variant effect prediction tools were quite accurate in identifying variants that altered DNA binding affinity, strikingly, none of the 42 tools we tested were able to accurately distinguish variants that altered DNA binding specificity. This observation is consistent with a prior study that found computational variant effect prediction tools to underperform on gain-of-function mutants as compared to loss-of-function mutants[71]. Existing variant effect prediction tools have not been trained on the effects of TF coding variants on DNA binding activity, pointing to the need for further development of such tools and expansion of such data sets both to identify the mechanisms of pathogenic variants and to serve as training data for improved variant effect prediction tools. Reduced DNA binding affinity may be predicted accurately if the variants diminish protein stability. Since many specificity-determining residues are variable across TFs, allowing

paralogous factors to recognize different DNA target sequences, variants that alter a TF's DNA binding specificity may be particularly difficult to discriminate.

We tested some identical missense mutations at the same position in members of the same or different HD subfamilies to explore the extent to which the entirety of the HD context of a variant influences its effects on affinity and/or specificity. In some cases we observed very different effects on DNA binding activity, expanding the context-specific effects beyond murine Hox, Nkx, Dlx, Lhx, Obox, and TALE HDs[16]. Strikingly, we observed such differences in effects even when comparing HDs from the same subfamily, which presents challenges for derivation of 'recognition rules' that generalize across HDs and also for variant effect prediction. Expanded mutational surveys of DBD subfamilies are needed to develop more accurate recognition models for TF-DNA binding and improved variant effect prediction tools.

Our results reveal that DNA backbone-contacting aa positions flanking core DNA recognition motifs can indirectly influence HD specificity for moderate affinity k-mers; mutations at aa residues flanking the core recognition motif have been found to decrease the DNA binding affinity of the C. elegans UNC-130 forkhead DNA binding domain and result in glial defects[72]. For example, the HOXD13 R306W (HD31) variant appears to differentially affect binding to CGTAAA- vs. AATAAA-containing 8-mers (Supplementary Fig. 7). In published crystal structures of the paralog HOXB13[73], Arg at the equivalent aa position makes backbone phosphate contacts with distinct configurations when bound to these different DNA sequences. This contact occurs in the flanking region outside the target DNA sequence, and in fact the sequence of the contacted flanking region is identical in the CGTAAA- vs. AATAAA-bound HOXB13 co-crystal structures. This kind of allosteric modification of a protein-DNA interaction suggests either that the alternate target sequences create subtly different flanking DNA shapes, one of which is better able to accommodate variation in the protein, or that the HD itself binds the alternate target sequences with a different geometry which entails different interactions with the flanking DNA. Differences in the DNA shape flanking recognition binding sites have been observed previously for S. cerevisiae basic-helix-loop-helix (bHLH) TFs[74].

Our results raise the question of why prior studies did not identify the individual HD positions that we discovered as contributing to DNA binding specificity (positions 4, 12, 16, 24, 25, 26, 31, 38, 40, 44, 46, 48, 53, 58), in particular since the major HD subfamilies that were the focus of our study (ANTP/HOXL, ANTP/NKL, PRD) were included in prior large-scale surveys of HD DNA binding specificities. Early studies that aimed to identify specificity-determining residues focused on DNA-contacting positions, and thus likely would have missed distal, non-DNA-contacting positions that contribute to DNA binding specificity, such as those we found. Similarly, a computational study that predicted HD binding motifs and inferred aa-base contacts was based on amino acid residues that contact DNA bases in available HD-DNA co-crystal structures[75]. Across TF structural classes, going beyond HDs, the potential contributions of distal, non-DNA-contacting positions towards DNA binding specificity have not been studied as much and have remained much less well understood. Here, we assayed variants at all but one HD position reported to be in close proximity to DNA, but that previously had not been identified as a specificity-determining position.

More recently, large-scale surveys of HD DNA binding specificity analyzed the reference alleles of HDs found in various species[7,8,68]; such surveys thus do not sample aa substitutions at highly conserved positions and thus are not able to determine their potential contributions to DNA binding specificity. In contrast, a key feature of our survey is that we assayed numerous rare variants, which resulted in our exploring a broader sequence space of variation at various HD positions than occurs when restricting to the reference alleles. For example, VENTX Y115F, which corresponds to HD position 25, showed altered specificity; Phe at

position 25 is infrequent in wildtype HDs, and was either absent[8] or rare[7,10] among HDs previously assayed in large-scale surveys of HD DNA binding specificity. Similarly, Gln and Trp are either rare or not found, respectively, at position 31 among previously surveyed HDs[7,8,10], and we found both of these substitutions to alter DNA binding specificity (VENTX R121Q, HOXD13 R306W). These results highlight how functional assay data of rare human TF coding variants empowered us to discover contributors to HD-DNA recognition. Furthermore, our data suggest that some of these positions might exhibit subfamily-specific or even individual HD-specific effects on DNA binding specificity, and thus their contributions to specificity might be missed in studies of other HDs.

The specificity-determining positions that we identified highlight the need to consider residues beyond those that contact DNA for their potential roles in DNA binding specificity. To our knowledge non-DNA-contacting residues have not been found previously to affect HD DNA binding specificity. DNA-distal residues have been found previously to influence the DNA binding specificity of the ETS domain TFs ELK-1 and SAP-1 by mediating the arrangement of residues in the DNA recognition helix with DNA[67]. Residues not contacting DNA in wing 2 of FoxN2, FoxN3, and FoxJ3 were found to affect DNA binding specificity of forkhead factors[76]. Similarly, distal residues of the yeast TF Pho4 were found in a large screen to affect DNA binding affinity by an unknown mechanism(s)[77]. Future studies are needed to determine the mechanisms by which distal residues affect DNA recognition and the potential roles of variation at such residues in the evolutionary diversification of TF families, transcriptional regulatory networks, and organismal phenotypes.

Our results highlight the need for functional assays of TF coding variants to reveal the potential pathogenic effects of clinical variants and to further elucidate mechanisms of TF-DNA recognition. 'Many-by-many' technologies need to be developed that will enable deep mutational scans of large libraries of TFs for their effects on DNA binding across a highly diverse library of DNA binding site sequences. We anticipate such expanded datasets will enable training of models to predict not only which TF variants are damaging, but also how they alter DNA binding activity.

## Methods

### Design of homeodomain constructs

**Identification of HD DBDs and classification into subfamilies.** HD DBDs were identified based on motif scanning using HMMER[78], with hmmscan version 3.2.1 (http://hmmer.org) used to identify TFs and DBDs that contained matches to the homeodomain Pfam HMM profile (PF00046 [https://www.ebi.ac.uk/interpro/entry/pfam/PF00046/]). The PF00046 seed alignment was visualized with WebLogo[79] for Fig. 1. HDs were classified into subfamilies (e.g., HOX/ANTP, HOX/NKL, SINE) based on published classifications[80]; PRD/PAXL and PRD/PAX subfamilies[80] were merged into a single PRD subfamily.

**Selection or design of variants for PBM assay.** HD missense variants were selected to span all HD DBD positions that were in close proximity to DNA but that had not been described previously in the literature as DNA specificity determining positions. We additionally included DBD missense variants that were annotated in the OMIM, ClinVar, or the UniProt databases as being disease-associated. For each position, we curated or designed variants to test the substitution in different HD subfamilies. We also selected or designed variants that were physicochemically conservative (e.g., substitution of a non-polar residue with another non-polar residue) versus non-physicochemically conservative to test the tolerance of particular HD positions to different types of amino acid substitutions. Where available, minor allele frequencies (MAFs) were obtained from the Genome Aggregation Database (gnomAD)[81], which was accessed on May 11, 2023 from https://registry.opendata.aws/broad-gnomad.

### Cloning and protein expression

**Cloning of wildtype and mutant HDs.** Reference TF DBD regions (TF DBD sequence plus 15 aa N- and C-terminal flanking sequences) were gene-synthesized (IDT gBlocks) and cloned into a Gateway compatible Entry vector (pDONR221), and then into an N-terminal GST protein fusion expression Destination vector (pDEST15). Mutant TF DBD sequences were generated by site-directed mutagenesis essentially as described previously[3] using primers shown in Supplementary Data 3. Cloned sequences (Supplementary Data 3) were full-length sequence-verified by Sanger sequencing.

**Protein expression.** N-terminal GST-fusion TF DBD proteins were expressed using the PURExpress in vitro transcription/translation kit (NEB) following the manufacturer's recommendations. Protein expression was verified and quantified by Western blot using recombinant GST protein (Sigma, G5663) as expression standards. Primary rabbit anti-GST polyclonal antibody (Sigma, G7781) (1:160,000) and secondary goat horseradish peroxidase-conjugated anti-rabbit IgG monoclonal antibody (Pierce, 31460) (1:200,000) were used for Western blotting. Mutant and reference alleles within allelic series were expressed and assayed by Western blotting at the same time.

### Protein binding microarray experiments

We employed an "all 10-mer" universal array design in 8 x 60 K, GSE format (Agilent Technologies; AMADID #030236). PBM double-stranding and subsequent experiments were performed following experimental procedures described previously[82,83]. All proteins were assayed at 200 nM final concentration in PBS (Supplementary Data 3). Allelic series were assayed in separate chambers on the same array. PBMs were scanned in a GenePix 4400 A microarray scanner. At least 3 scans were taken for each slide at different photomultiplier tube (PMT) gain settings.

### Statistical analysis of PBM data

**Array image quantification and data normalization.** We refer to the set of array chambers for a single allelic series assayed on the same slide (up to 8 array chambers per slide) as an allelic replicate. For each allelic replicate, an optimal scan was selected, corresponding to the PMT gain with the lowest proportions of both over-saturated probes (over-saturated probes are defined here as having intensity levels of $>2^{15.5}$; the signal is completely saturated at $2^{16} = 65,536$ intensity levels) and under-saturated probes (defined as intensity levels $<2^5$) across all chambers in the allelic replicate. Most often, this corresponded to PMT gains between 400 and 500. The same PMT gain value was typically used within an allelic series.

Pre-processing procedures, including background subtraction, and spatial de-biasing, were applied to each array independently as described previously using the PBM Analysis Suite[82,83]. Subsequent normalizations, scoring of 8-mers, and statistical analyses were performed within the upbm data analysis pipeline that we developed, using the functions described below. Probes with foreground intensities less than background intensities were excluded. This was performed using the 'backgroundSubtract' and 'spatiallyAdjust' functions with default parameters. An empirical reference generated by pooling across Cy3 scans from 90 arrays was used for Cy3 normalization and is available in the upbmAux R data package. Probes with $\log_2$ Cy3 observed-to-expected ratios greater than 1 or less than −1 were also excluded. This was performed using the 'cy3FitEmpirical' function with threshold = 1 and the 'cy3Normalize' function with default parameters.

Arrays were normalized within allelic replicates against a single anchor sample within the replicate, typically the reference allele. A log-scale additive normalization factor was estimated for each non-anchor sample using the trimmed mean of M-values approach originally proposed for RNA-seq normalization[84]. This was performed using the 'normalizeWithinReplicates' function with default parameters.

Arrays were then normalized across allelic replicates using a single anchor allele present in all allelic replicates considered, which was typically the reference allele, with the exception of two allelic series (HOXD13 and SIX6) for which we used multiple anchor variant alleles (and averaged normalization factors across these multiple anchors). First, a set of cross-replicate reference sample values was calculated for the anchor allele using the log-scale mean of the anchor samples across the replicates. Then, to account for both shift and scale differences between allelic replicates, both a log-scale additive factor and a log-scale multiplicative factor were calculated for each replicate against the cross-replicate reference (normalization) sample. The log-scale additive factor for each replicate was estimated as the difference in the log-scale median probe intensity of the corresponding anchor sample and log-scale median probe intensity of the set of cross-replicate reference sample values. The log-scale multiplicative factor for each replicate was estimated as the median ratio of the rank-ordered and median-centered log-probe intensities between the corresponding anchor sample and the set of cross-replicate reference sample values. This is approximately equivalent to the slope of the log-scale QQ plot of the anchor and cross-replicate reference samples. This was performed using the 'normalizeAcrossReplicates' function with default parameters.

Allelic replicates with low relative signal (as compared to other replicates) were deemed as lower-quality experiments and therefore filtered out at this step. This was assessed by comparing the upper-tail width ($90^{th}$ to $99^{th}$ percentile) of the distribution of probe intensities for reference TF DBD samples in each allelic replicate. In most cases, at least two replicates were assayed for each member of an allelic series; for a small number of published data[3] reanalyzed for inclusion in this data set, a single variant array was analyzed by comparison with replicates of the reference allele. If the tail width of a reference TF sample in an allelic replicate was less than 25% of the maximum tail width (or 50% of the maximum, if the array had been used previously and then stripped for re-use of the array in a second PBM experiment) of the reference TF DBD samples across all replicates of the allelic series, the entire allelic replicate was excluded due to it being deemed as a lower quality replicate. Normalization across allelic replicates was repeated as above after removing the lower-quality experiments.

**Scoring of 8-mers.** After cross-replicate normalization, mixed-effects models were fitted for each 8-mer aggregating across allelic replicates and probes to calculate 8-mer-level affinity scores. Summarization was performed for each 8-mer by first estimating probe-level means and variances across replicates using the limma smoothed variance estimator[85,86]. This was performed using the 'probeFit' function with guardrail = FALSE. Each probe-level estimate was further corrected for sequence position bias to account for differences in measured binding due to the distance of the 8-mer from the beginning or end of the probe sequence. Finally, 8-mer-level affinity scores were calculated using the two-step DerSimonian and Laird estimator across all probes containing the respective 8-mer sequences[87]. All inferences were then performed at the 8-mer-level. This was performed using the 'kmerFit' function with default parameters.

Comparisons of 8-mer affinity estimates across proteins were visualized as contrast-versus-reference plots: the difference in binding affinity per 8-mer between two alleles of interest (variant minus reference) is plotted on the $y$-axis, and the binding affinity per 8-mer of the reference allele is plotted on the $x$-axis.

**Identification of preferentially bound 8-mers.** For each TF DBD allele, 8-mer preferential affinity was calculated based on a normal-exponential convolution model fitted across 8-mers[88]. The estimated exponential component for each 8-mer was tested against a null hypothesis of zero, yielding a preferential affinity effect size, normal approximation-based p-value, and the Benjamini-Hochberg corrected statistical significance ($q_{Aff}$, referred to as "affinityQ") for binding to

each 8-mer. This was performed using the 'kmerTestAffinity' function with default parameters.

**Identification of differentially bound 8-mers.** Differential affinity was tested for each 8-mer between a reference and variant TF DBD allele pair by testing for a difference in the 8-mer-level affinity scores against a null hypothesis of zero. Since 8-mer-level affinity scores are estimated for each TF DBD using the same set of probes, we do not assume the covariance of the two affinity scores to be zero. Instead, we estimate the covariance across TF DBD pairs for each 8-mer using the empirical covariance of the probe-level means and the random probe-effect variance estimates from the two-step DerSimonian and Laird estimator as applied above. For each 8-mer, we report the differential effect size calculated by subtracting the corresponding reference allele affinity score from the corresponding variant's affinity score, a normal approximation-based $p$-value, and the Benjamini-Hochberg corrected statistical significance ($q_{Diff}$, referred to as "contrastQ") of the difference. This was performed using the 'kmerTestContrast' function with default parameters.

**Identification of differentially specific 8-mers.** Differential specificity was tested for each 8-mer between a reference and a variant TF DBD allele pair by first fitting a cross-8-mer differential trend to the corresponding MA plot with reference allele affinities on the x-axis. The global trend was fitted using weighted ordinary linear regression with a B-splines basis. Two trends were fitted to each MA plot to capture any differential specificities that may be smoothed out by fitting a single trend to the data. The two trends were identified by first splitting the data along the $x$-axis into 20 equal-width bins and fitting a two-component univariate normal mixture to the $y$-axis values in each bin. The relative upper and lower components in each clustering were then grouped across the bins. The two trends were fitted separately by weighing each $k$-mer based on the likelihood of belonging to either the upper or lower clusters. Finally, a single trend was selected to be used for testing by taking the trend which minimized the mean residual across the top 50% of 8-mers.

We considered an 8-mer as displaying a specificity change between variant and reference TF DBD alleles when the differences in the measured affinity scores deviated from the trend of a global change in affinity. For each 8-mer, we report the specificity effect size calculated by subtracting the differential effect size from the trend line, a normal approximation-based p-value, and the Benjamini-Hochberg corrected statistical significance ($q_{Spec}$, referred to as "specificityQ") of the difference from the trend. This was performed using the 'kmerTestSpecificity' function with default parameters.

**Altered DNA-binding affinity criteria.** We called a variant as showing altered DNA-binding affinity if more than 15% of the 8-mers which were preferential ("preferential affinity" 8-mers) in the reference TF allele (i.e. with $q_{Aff}$ values $<1 \times 10^{-6}$) are differential, with contrastQ values $<1 \times 10^{-6}$, in one direction (gain of binding affinity or loss of binding affinity). This threshold was defined based on a 5% FDR, using an empirical null distribution constructed from the null comparisons of HD reference alleles from at least quadruplicate replicate PBM experiments, in which we compared TF DBD reference alleles against replicate data of the same protein.

The null comparisons were performed using reference alleles from 4 ALX3, 4 HOXA4, 4 MSX2, 4 NKX2-4, 8 PITX2, and 4 VENTX replicate PBM experiments. For each null comparison, the reference alleles for one TF were permuted to reference and non-reference labels to produce 6, 6, 6, 6, 20, and 6 null comparisons, respectively, for a total of 50 null comparisons.

**Altered DNA-binding specificity criteria.** We called a variant as showing altered DNA-binding specificity, if the variant had a minimum

of 10 8-mers with affinityQ values $< 1 \times 10^{-6}$ and specificityQ values $< 1 \times 10^{-6}$. This threshold was defined based on a 5% FDR, using an empirical null distribution constructed from the null comparisons of HD reference alleles as described above, in which we compared reference alleles against replicate data of the same protein.

To complement this criterion, especially to give greater confidence in interpreting variant alleles which also showed reduced DNA-binding affinity or more subtle DNA-binding changes, we added a related criterion. This additional criterion takes into account the assumption that 8-mer preferences that are affected by an alteration in TF DNA-binding specificity would share some DNA sequence similarity and reflect a common motif core among those 8-mers. Similar to the other DNA-binding changes criteria, we used empirical null distributions constructed from the null comparisons of HD reference alleles, in which we compared TF DBD reference alleles against replicate data of the same protein. Thresholds were set at 5% FDR, using null distributions constructed from the reference TF null comparisons. We considered the edit distance of all 8-mers in variant allele versus reference allele comparisons (or reference allele versus reference allele comparisons, for the null comparisons) against a seed 8-mer, which we defined as the 8-mer with the maximal magnitude of contrastResidual value, either amongst the 8-mers with negative contrastResidual values, or the 8-mers with positive contrastResidual values. The edit distance metric we used is a Levenshtein distance with equal weights (unit costs) for substitutions, insertions, and deletions, and the use of this edit distance is predicated on the assumption that there is likely some kind of shared, core motif amongst subsets of 8-mers in cases of altered TF-DNA binding specificity. The contrastResidual value for each 8-mer was calculated based on the deviation of the measured difference in the binding affinity between the variant TF allele of interest and the reference TF allele for an 8-mer, against the B-spline fits in the differential specificity analyses.

Subsequent to analysis with the upbm data analysis pipeline, we counted the numbers of 8-mers that met a particular edit distance cut-off value against the seed 8-mer (e.g., edit distance <2 against the seed 8-mer). We identified variants that had a supra-threshold number of 8-mers for combinations of specificityQ and edit distance cut-off values, with variants needing to meet a panel of thresholds (multiple combinations of specificityQ and edit distance cut-off values, e.g., minimally edit distance <2 by specificityQ <0.05 and specificityQ <0.01, or edit distance <3 by specificityQ <0.05 and specificityQ <0.01; Supplementary Fig. 8), given the more relaxed specificityQ cut-off values. We called these variants as demonstrating altered DNA-binding specificity. For example, the 5% FDR in the empirical null distribution corresponded to a null comparison having eight 8-mers that had an edit distance of <2 to the seed 8-mer (8-mer with the maximal magnitude of contrastResidual value); these eight 8-mers also had specificityQ values < 0.05. Variants with numbers of 8-mers meeting the criteria (in this example, edit distance <2 to seed 8-mer, and specificityQ <0.05) that were above this 5% FDR threshold were considered as variants with altered DNA-binding specificity. The use of a panel of thresholds (multiple specificityQ per edit distance cut-off value) increased our confidence in the altered DNA-binding specificity calls, especially since there are fewer 8-mers within the null distributions at the shorter edit distance cut-off values.

We considered 8-mers with negative contrastResidual values separately from 8-mers with positive contrastResidual values (and likewise, for the null distributions), since the biological interpretation may be different for these two types of 8-mers (e.g., selective loss of DNA-binding specificity, versus recognition of previously uncharacterized sets of 8-mers / retention of binding to 8-mers that are most strongly bound by the reference TF, respectively).

We manually reviewed the differential specificity MA plots and removed one variant call identified by the edit distance criteria

(ARX R332H) which appeared to be a false positive due to the B-spline fit for this specific variant.

## Analysis of variant effect prediction tools

We used dbNSFP v4.3[89] to obtain precomputed pathogenicity scores from 40 predictors including conservation scores from 5 sources (Supplementary Data 5). We utilized converted rankscores where necessary to ensure that the direction of scores was monotonic, meaning that higher scores indicate a higher likelihood of pathogenicity. We also included pathogenicity scores from EVE[90], a deep generative model that captures evolutionary constraints on protein sequences, and AlphaMissense, a model based on AlphaFold[91] that combines structural and evolutionary constraints with machine learning of protein sequence probabilities[43]. Variants with missing predictions were counted as false negatives or assigned a prediction value of 0. The pathogenicity scores from various variant effect predictors could be obtained for 80 of the 92 assayed variants, and we evaluated these scores against our altered DNA binding affinity/specificity calls.

## Homology modeling

We employed a comparative modeling approach using RosettaCM[92] to model structures of TFs in this study with unknown structural information. With the exception of NKX2-5 (PDB: 3RKQ), all TFs were reliably modeled using TF templates that share at least 40% sequence identity (Supplementary Table 2), and that are co-crystallized with DNA. Wherever feasible, the templates were selected from within the same HD subfamily. We designed the protein-DNA interface in all the structures using the top scoring 8-mer sequences obtained in our experiments for each reference allele. We used the DnaInterfacePacker mover from Rosetta suite for this purpose, with an energy function optimized for protein-DNA interactions[93]. All structural models along with their corresponding DNA 8-mers were energy minimized and refined using the FastRelax protocol from the Rosetta suite.

## ChIP-Seq data analysis

For CRX ChIP-seq analysis, ChIP-seq (SRR10172865 [https://www.ncbi.nlm.nih.gov/sra/?term=SRR10172865]) and the corresponding input (SRR10172850 [https://www.ncbi.nlm.nih.gov/sra/?term=SRR10172850]) reads were downloaded from the Sequence Read Archive (SRA) and mapped to version GRCh37 (hg19) of the human genome with Bowtie 2[94]. Narrow peaks were called using MACS2[95] with default parameters, and windows of 150 bp centered on the peak summits for the top 1000 peaks (sorted by $-\log_{10}$ P-value) were used as the foreground set. For SIX6 ChIP-seq analysis, ChIP-seq (SRR7817899 [https://www.ncbi.nlm.nih.gov/sra/?term=SRR7817899]) and the corresponding input (SRR7817908 [https://www.ncbi.nlm.nih.gov/sra/?term=SRR7817908]) reads were downloaded from the SRA and mapped to version GRCm38 (mm10) of the mouse genome with Bowtie 2. Narrow peaks were called using MACS2 with paired-end default parameters, and windows of 150 bp centered on the peak summits for the top 1000 peaks (sorted by ChIP enrichment score) were used as the foreground set. For each peak set, a genomic background matched for length, G/C content, CpG content, promoter overlap, and repeat content was generated using GENRE[61] applied to the mouse genome[96]. Foreground and background regions containing a given 6-mer were counted and the significance of enrichment was tested by Fisher's Exact Test in R. Sex of the donor was not considered when curating available ChIP data; both samples (human donor retina and mouse embryonic fibroblast cell line NIH-3T3) were from male donors.

## Reporting summary

Further information on research design is available in the Nature Portfolio Reporting Summary linked to this article.

## Data availability

PBM data have been deposited in the GEO database under accession number GSE233827. Supplementary Data 4 and the Source Data file for images are available from the Harvard Dataverse repository https://doi.org/10.7910/DVN/FDQHCF[37].

## Code availability

The source code for the upbm pipeline and statistical framework is available at https://github.com/pkimes/upbm[99].

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

## Acknowledgements

We thank Luca Mariani and Katy Weinand for helpful discussions and Kaia Mattioli and Shubham Khetan for critical reading of the manuscript. This work was supported by the National Institutes of Health (grants R01 HG010501 to M.L.B. and R35 CA220340 to S.C.B.), an A*STAR National Science Scholarship to K.H.K., and a National Science Foundation Graduate Research Fellowship to J.M.R.; J.M.R. was also supported by 5T32HG002295-13.

## Author contributions

M.L.B. conceived the research project. M.L.B., K.H.K, P.K., S.I. and J.M.R. designed the research project. S.I., S.K.P., J.A., J.V.K. and J.M.R. performed experiments. K.H.K., P.K., S.I., D.S., G.R., C.L., S.S.G., S.K.P., J.A., J.V.K., J.M.R. and R.J. performed analyses. K.H.K., P.K., S.S.G. and G.R. prepared the figures. M.L.B., S.B. and R.I. supervised the research. K.H.K., P.K., S.S.G. and M.L.B. wrote the manuscript. All authors approved the final version of the manuscript.

## Competing interests

M.L.B. is a co-inventor on U.S. patents # 6,548,021 and #8,530,638 on PBM technology and corresponding universal sequence designs, respectively, which were used to assay the DNA binding activities of the HD proteins in this study. The remaining authors declare no competing interests. Universal PBM array designs used in this study are available via a Materials Transfer Agreement with The Brigham & Women's Hospital, Inc.

## Additional information

[1]Division of Genetics, Department of Medicine, Brigham and Women's Hospital and Harvard Medical School, Boston, USA. [2]Program in Biological and Biomedical Sciences, Harvard University, Cambridge, MA, USA. [3]Department of Biostatistics and Computational Biology, Dana-Farber Cancer Institute, Boston, MA, USA. [4]Department of Biostatistics, Harvard T.H. Chan School of Public Health, Boston, MA, USA. [5]Boston Bangalore Biosciences Beginnings Program, Harvard University, Cambridge, MA, USA. [6]Department of Biological Chemistry and Molecular Pharmacology, Blavatnik Institute, Harvard Medical

School, Boston, MA, USA. [7]Department of Cancer Biology, Dana Farber Cancer Institute, Boston, MA, USA. [8]Committee on Higher Degrees in Biophysics, Harvard University, Cambridge, MA, USA. [9]Bioinformatics and Integrative Genomics Graduate Program, Harvard University, Cambridge, MA, USA. [10]Department of Data Science, Dana-Farber Cancer Institute, Boston, MA, USA. [11]Department of Pathology, Brigham and Women's Hospital and Harvard Medical School, Boston, MA, USA. [12]These authors contributed equally: Kian Hong Kock, Patrick K. Kimes, Stephen S. Gisselbrecht.
✉e-mail: mlbulyk@genetics.med.harvard.edu

