## [Peer Review File · Nature Communications]

DNA binding analysis of rare variants in homeodomains reveals previously unknown homeodomain specificity-determining residuesREVIEWER COMMENTS

Reviewer #1 (Remarks to the Author):

Authors present a well-written paper with extensive and detailed Methods. They studied the effect in vitro of missense variants in each of the homeodomain amino acid positions on DNA-binding affinity and/or specificity and showed a range of effects, such as increased affinity for low-affinity (w.r.t. wild-type) binding sites and affinity changes when some non-nucleotide-contact residues are mutated. They also presented the (lack of) accuracy of numerous variant effect prediction tools with regards to affinity and specificity changes. The detailed analysis in this paper could be used to feed back into the development of such algorithms to improve, particularly, their specificity-change predictions.

In the annotated version of the manuscript I have remarked on some suggested style changes. It is generally advised to spell out numbers up to ten (AP, NYT, APA) (or twenty, or even one hundred according to the Chicago Manual of Style) when they are not associated with data (units, coordinates, dates, etc.) and always at the sentence start. Also, most semi-colons (;) can probably be replaced with periods (.), and for genome builds the official nomenclature (e.g. GRCh37) should be used, with synonym (e.g. hg19) in brackets. Most instances of 'damaged DNA binding' (or similar) can probably be replaced with 'altered DNA binding', since enhanced affinity and changed specificity are not 'damage' (even if the consequences are damaging to the individual).

Is there any significance to the fact that among the alleles shown in supplemental Fig.4, 28 (of 92) involve an arginine (eight of which are R-to-Q)? Across Figures 3+4+5 it is 10/16 (6 R->Q) and Table 1 it's 11/17 (4 R->Q). I haven't checked whether the different R's from different proteins represent the same core HD amino acid (e.g. whether SHOX-R173 = NKX2-5-R139). Maybe it's partially explained by the fact that in the canonical HD, at six positions R is the most prevalent and at at least five positions the second-most prevalent (Fig.1a). And, related to that, that a disproportionate number of those arginines are mutated to glutamine (when single point nucleotide mutations can change the six R codons into a combined 13 different codons)? Are arginine mutations more damaging? Are R-to-Q mutations more damaging than other R mutations? Or the other way around (i.e. you don't see other mutations because they are more damaging and embryonic/neonatal lethal)?

All-in-all I have no problem recommending this paper for publication.

Reviewer #2 (Remarks to the Author):

Summary: In this paper, Kock et al use protein binding microarrays to assess how a large number of variants in human homeodomain (HD) proteins impact DNA binding affinity and specificity. To analyze the data, the authors developed new approaches to assess and visualize changes in binding to individual 8-mer sequences. Overall, the experiments and analysis are well performed, and the data support the idea that numerous different residues outside the classically defined DNA binding contact amino acid positions can influence both DNA binding affinity and/or specificity. These data serve as a valuable resource for those studying HD proteins. In addition, the data reveals that our current computational methods to predict how specific variants impact DNA specificity are insufficient to accurately assess changes in DNA binding and that additional empirical experiments coupled with predictive tools are needed to bridge this knowledge gap. For example, while the authors do identify new residue positions in the 60 amino acid HD that can impact DNA binding, they further found that other variants in these same positions can result in either no change in binding, an affinity change, a specificity change, or a novel specificity change. Hence, generalized rules for how variants in key amino acid positions impact DNA binding are not established in this study. Nevertheless, revealing this knowledge gap is valuable to the field and the authors provide an experimental platform and new way to analyze PBM data to highlight changes in DNA binding to 8mer sequences.

Below are a list of comments that should be addressed to both increase the ease of understanding the data and to better validate the results:

- 1) The authors list all tested factors by name and variant within the context of the full-length protein such as ALX3 L168V. However, it would also be helpful to include a reference to where the amino acid variant is located within the 60 amino acid HD such as ALX3 L168V (HD#XX) in both the Figures and the text. This would make it much easier to determine where the variant is within the HD and if it impacts the N-terminal ARM, Helix1, 2, or 3, etc without referring to Supplementary table 2.
- 2) The guide graph in Figure 3A is very helpful. However, in Figure 4, the X-axis changes from contrastAverage to reference affinity. Are these the same? If so, consistent nomenclature should be used. If not, please consider creating a guide like in Figure 3A within Figure 4 to highlight what the "reference affinity" axis is and how it differs from contrast Average.
- 3) Since the analysis of the PBM data is novel (upbm), it would be helpful to provide a more thorough explanation of the analysis. In addition to point #2 above, the following are suggestions:
 - a. Supplementary figure 2 summarizes the key features of upbm. Consider incorporating this diagram or an abbreviated version of it into Figure 2.
 - b. Explain whether the numbers along the X-axis of each graph are relative and therefore apply to just that control/variant or absolute and can be compared across different graphs/factors/variants. In other words, what is upbm affinity? Is it an absolute measurement or used primarily for relative comparisons?
 - c. I became a little confused as to how to visualize the graphs and determine if a variant is selectively impacting the affinity to a subset of 8mers or generally impacting the affinity to all sites. Hence, it would be helpful if the authors showed clear example graphs of representative variants that : i) strongly disrupt all DNA binding compared to WT; ii) Weakly disrupt DNA binding affinity to all 8mers; versus iii) Disrupt DNA binding affinity to only a subset of 8mers. These examples would help the readers interpret the graphs in a more intuitive manner.
- 4) Of the datasets that were reprocessed with the new upbm pipeline, it would be informative to include which alleles yielded different results from previous analysis/studies.
- 5) In ChIP analyses, I am assuming that FG stands for foreground and BG stands for background, but I do not see a place where this is defined. Also, were only the top 1000 peaks used for analysis? If so, this needs to be mentioned in the main text. Lastly, it is unclear why the authors haven't used common, validated, and accepted methods in the field for this such as HOMER, MEME, or FIMO. I understand the authors wanted to search for only one sequence instead of a PWM but as far as I'm aware this is possible with the forementioned methods. These tools would apply to both human and mouse genomes whereas the mentioned method GENRE does not currently apply to mouse genomes. Further, why were different peak windows used in the CRX (200 bp) and SIX (80bp) ChIP-seq analyses?
- 6) The ability of upbm to detect a gain of specificity and/or affinity should be validated with another DNA binding method.
- 7) Better explanation of the possible association of VUS alleles and disease is warranted. For example, while some variants tested in the paper were previously shown to be pathogenic, others were VUSs. In some sections of the paper, the authors make associations between their data and VUSs that are not well justified or need to be better justified based on available genetic data. A prime example is the finding that the PBX4 rare variant R215Q results in a severe loss of affinity. The authors subsequently argue that this variant may underly hematological and metabolic defects. But it is unclear if this variant was found in a patient or patients with such defects, and whether the patient(s) are heterozygous or homozygous for such variants. The authors should better integrate available genomic data that provides insight into whether VUS alleles of factors such as PBX4 are likely haploinsufficient or not. In scanning the gnomad database, PBX4 has a PLI of 0 – suggesting PBX4 is not haploinsufficient. Hence any argument that the R215Q allele is associated with disease would either require evidence of homozygous alleles in humans or a possible explanation of how the R215Q functions as a potential dominant negative allele. Hence, the human allele data, patient symptom data, and the PBM binding data using different alleles needs to be better explained and clarified.
- 8) The difference between yes/- and yes/+ in Supplementary Table 2 is unclear.
- 9) What does replicate refer to in Supplementary Table 3? Is it the number of total replicates

performed on each variant? If so, what is the justification between the different replicate numbers?

10) Unclear where the following data is shown: "Our homology modeling results suggest that SIX6 Thr-165 at position 38 contacts Thr-139 at position 12, Leu-143 at position 16, and Leu-167 at position 40; intriguingly, these residues are within, or adjacent to, the hydrophobic core, and thus are not ones that would have been expected a priori to contribute to DNA binding specificity."

11) It would benefit the reader to show images of the crystal structure of some of the key residues (positions 58, 44, 25, 26, and 31) the authors are highlighting starting on page 14.

12) While there are many panels in Supplementary Figure 4, adding labels such as A, B, C for each factor would make it much easier to find the referenced data throughout the paper without having to look through all the pages of graphs.

Minor comments:

- Second half of the first paragraph of the introduction is the same statement word for word as the beginning of the abstract.
- It would be helpful for the reader to label every fifth residue (or more) in Figure 1b.
- Please list the four affinity-altering variants on page 10.
- Top of page 16, there should be a comma between 12 and ALX4 Q225E

Reviewer #3 (Remarks to the Author):

The manuscript entitled "DNA binding analysis of rare variants in homeodomains reveals novel homeodomain specificity-determining residues" presents a survey of DNA binding affinity and specificity profiles for 92 variants of homeodomain transcription factors. This study utilizes protein-binding microarrays (PBMs) which are analyzed using a modified pipeline ("upbm") in order to assess binding to both high- and lower-affinity 8-mers and to better observe differences in specificity. As a large TF family with many disease associations, homeodomains are of significant academic and clinical relevance. Furthermore, variants tested include pathogenic and likely pathogenic variants associated with a variety of diseases. The authors identify 14 novel residues within homeodomain regions that affect DNA binding affinity or specificity. Of these 14 residues, five are not thought to contact DNA directly, a key result that supports recent findings that changes outside of DNA-contacting residues or even outside the borders of DNA binding domains can affect DNA binding of TFs. For many known and novel specificity-determining residues, the authors observe that lower-affinity sites, rather than the highest affinity sites, are often differentially bound by homeodomain variants. Importantly, this finding provides evidence for the emerging understanding that DNA occupancy of TFs is largely determined by the composite effect of several lower-affinity interactions. The authors highlight that DNA binding prediction tools are especially poor for predicting differences in specificity, illustrating the need for experimental studies such as this one. To this point, the authors observe that seemingly analogous mutations in different transcription factors can have distinct effects on DNA binding. Importantly, the authors support their findings using structural models to generate hypotheses about physiochemical interactions that may be involved in differential DNA binding.

Overall, this manuscript is very clear and well-written, facilitating easy understanding of the results. The findings are of significant interest to the transcription factor field, as they illustrate properties of TF-DNA binding that are of emerging importance and cannot currently be predicted using in silico approaches. The authors' findings that mutations in homeodomain regions can have such distinct effects across different transcription factors suggest that similar or more marked differences may also exist within TF families with more variable DNA binding domains. Considering the clinical relevance of the mutations studied, the manuscript also provides valuable insight into mechanisms connecting rare variants and their associated clinical phenotypes. Due to all these reasons, I recommend publication of the article following some minor revisions suggested below to improve clarity and the precision of stated conclusions.

- 1) It would be useful to add the frequency of the variants tested in Table S2.
- 2) Introduction, paragraph 4: "upm" should be "upbm."

- 3) In Fig. 1 it is somewhat difficult to visualize which residues correspond to which bars as the 3 panels are difficult to align. Consider using a heatmap with a layout similar to Fig. 5a.
- 4) More information should be reported in the Results section and maybe represented in a main figure regarding identity of the 92 variants tested, namely the number of distinct TFs represented per TF and subfamily and the number of variants of each type (pathogenic, likely pathogenic, VUS). Although this information is present in Table S2, it may be convenient for the reader to have it available in the text and figure.
- 5) Supp. Fig. 3: This figure seems to be misnumbered and missing panels referenced in the text. Panels referring to HOXD3 are listed as Supp. Fig. 3j-m in the text but are shown in panels b-e in the figure. Panels n,o in the text should be f and g (these panels are also hard to read in the figure). Relatedly, The text mentions a comparison between Msn2 and Msn4 in Supplementary Fig. 3 b-e but these are not included in the figure; additionally, Msn4 is not mentioned earlier in the text along with Msn2, Com2, and Usv1 as a TF that was studied, so it's unclear where Msn4 comes into play.
- 6) Fig. 2a: Consider replacing this pie chart with something more intuitive to help the reader more easily interpret the result. Since two variables are being considered (affinity and specificity), it may help to represent these variables separately (e.g., as a contingency table, heatmap, or similar). This is just a recommendation and it's up to the authors whether changes are made.
- 7) Fig. 2b-d: Since the less successful prediction tools are discussed in the main text, the reader may expect these to also be represented in the main figure panels. While the figure caption explains that only the top 10 most successful tools are shown, it may help to explain this in the text as well to avoid confusion. Alternatively, consider the benefits of including results for all tools (as in Supp. Fig. 5) in the main figure, perhaps with the top 10 best tools with thicker lines and the remaining ones with thin lines. This is just a recommendation and it's up to the authors whether changes are made.
- 8) Fig. 3: Throughout the data panels b-g displaying contrastDifference and contrastAverage, it is somewhat easy to identify decreased affinity, but changes in specificity are harder to observe for some examples mentioned. For example, it is difficult to see why ALX3 L168V reduces affinity while HOXA4 R217L reduces affinity and also changes specificity when the plots look somewhat similar. It might help to highlight the regions in the real case plots that show difference in specificity.
- 9) P10 top paragraph: "Finally, 16 of 32 missense variants that were present only in gnomAD..." It would be helpful to be more explicit about how you are interpreting a variant that is only present in gnomAD (i.e., that are common variants and/or have no clinical association, or if rare they may affect function).
- 10) P10 bottom paragraph: This paragraph talks extensively about structural details, including mentions of a hydrophobic core and a helix 1 proline substitution. It may help to use structural figures/models here to introduce and help the reader to visualize these important structural features.
- 11) Fig. 3g: "partial sparing of the highest-affinity 8-mers" by HESX1 R109Q is not apparent to the inexperienced reader from looking at the figure. Consider some indications in the figure to help illustrate this point.
- 12) Fig. 4a: Consider also highlighting "retained high-affinity 8-mers" that are mentioned in the results text, as they are not apparent in the figure.
- 13) P11: "suggesting... that this variant may also lead to cranial structural birth defects" seems like an overstatement as this study does not identify molecular or phenotypic outcomes of this variant. Similarly, with the NKX2-6 R150C variant in P10. Consider toning this down, or moving the discussion of potential disease relevance to the Discussion section.
- 14) Fig. 4c,g,l: Enrichment of motifs within ChIP-seq peaks is not easily interpreted from these graphs. Consider alternative representations, e.g. by representing proportions. Also, define "FG" and "BG" in the figure and caption.
- 15) Fig. 4e-g: It is not clear if or how ChIP-seq binding of wildtype SIX6 to the noncanonical TCACAC site suggests that "SIX6 T165A may exhibit enhanced binding at these sites in vivo, and that gain of specificity by SIX6 variants may be pathogenic." Consider alternative and toned-down interpretations. Additionally, it is unclear why the T165G mutation was also tested and what conclusions can be drawn from it.
- 16) In Fig. 3, PBM results are represented with "contrastAverage" as the X-axis variable. However, Figs. 4 and 5 use "reference affinity" as the X-axis variable. It's not clear why this variable is not

consistent across all figures or if this changes how readers should interpret the panels in Fig. 3 vs. Figs. 4 and 5.

17) Fig. 4h,i: The results text states that VENTX R121Q and Y115F show preferential loss of binding to lower-affinity sites, but this is not clear to the average reader in the figure panels, where the highlighted 8-mers appear to correspond to higher-affinity sites on the far right side of the graph. If possible, this should somehow be made more clear to the reader.

18) P13, top paragraph: "...suggesting that HD mutants that perturb binding to lower affinity sites may affect gene regulation" may be an overstatement as the data mentioned only pertain to TF-DNA binding and not to recruitment of cofactors/transcriptional machinery or changes in transcription.

19) P14 bottom paragraph: "...suggesting indirect readout of the underlying DNA sequence"; it is not clear what is meant by "indirect readout" or how the PBM results suggest this.

20) P20, top paragraph: Here the authors mention that a main goal was "...to explore potential context-specific effects" of a mutation in the same residue in different TFs. Consider commenting on/reminding the reader about what context features may affect the outcome of a mutation.

21) Fig. 5a "current study variants": Consider as a heat map to show the proportion of variants at each position that display each type of change (alternatively, add the number in each cell). Also, consider using a different color (grey) to indicate positions where no variant was tested, so the reader can differentiate between residues not tested and residues that did not appear to have an effect.

Juan Fuxman Bass

Point-by-point response to reviewer comments on manuscript NCOMMS-23-38131-T "DNA binding analysis of rare variants in homeodomains reveals novel specificity-determining residues":

Reviewer #1 (Remarks to the Author):

Authors present a well-written paper with extensive and detailed Methods. They studied the effect in vitro of missense variants in each of the homeodomain amino acid positions on DNA-binding affinity and/or specificity and showed a range of effects, such as increased affinity for low-affinity (w.r.t. wild-type) binding sites and affinity changes when some non-nucleotide-contact residues are mutated. They also presented the (lack of) accuracy of numerous variant effect prediction tools with regards to affinity and specificity changes. The detailed analysis in this paper could be used to feed back into the development of such algorithms to improve, particularly, their specificity-change predictions.

In the annotated version of the manuscript I have remarked on some suggested style changes. It is generally advised to spell out numbers up to ten (AP, NYT, APA) (or twenty, or even one hundred according to the Chicago Manual of Style) when they are not associated with data (units, coordinates, dates, etc.) and always at the sentence start. Also, most semi-colons (;) can probably be replaced with periods (.), and for genome builds the official nomenclature (e.g. GRCh37) should be used, with synonym (e.g. hg19) in brackets. Most instances of 'damaged DNA binding' (or similar) can probably be replaced with 'altered DNA binding', since enhanced affinity and changed specificity are not 'damage' (even if the consequences are damaging to the individual).

We have spelled out all sentence-initial numbers. We disagree that most of the highlighted single-digit numbers are not "associated with data." The units here are variants tested, and in our opinion changing the numerals to words decreases the readability of results. We agree that "damaged" DNA binding is a more subjective term than "altered" and have changed the text accordingly, and have replaced the ambiguous "respectively" with a more explicit elaboration of the modeling results.

Is there any significance to the fact that among the alleles shown in supplemental Fig.4, 28 (of 92) involve an arginine (eight of which are R-to-Q)? Across Figures 3+4+5 it is 10/16 (6 R->Q) and Table 1 it's 11/17 (4 R->Q). I haven't checked whether the different R's from different proteins represent the same core HD amino acid (e.g. whether SHOX-R173 = NKX2-5-R139). Maybe it's partially explained by the fact that in the canonical HD, at six positions R is the most prevalent and at at least five positions the second-most prevalent (Fig.1a). And, related to that, that a disproportionate number of those arginines are mutated to glutamine (when single point nucleotide mutations can change the six R codons into a combined 13 different codons)? Are arginine mutations more damaging? Are R-to-Q mutations more damaging than other R mutations? Or the other way around (i.e. you don't see other mutations because they are more damaging and embryonic/neonatal lethal)?

Like other DNA-binding domains, the homeodomain is enriched in positively charged amino acids to facilitate binding to the negatively charged DNA. There is a particular propensity to use arginine residues to recognize minor groove width features as well as for some base-specific contacts, which probably explains its overrepresentation in the dataset. Several of the R-to-Q variants tested were chosen to address how the effects of an identical change at a corresponding position depend on context (see Figure 6c-e and the discussion thereof), which probably contributed to the enrichment of that particular change. Our data do not suggest that R-to-Q mutations are special in this way—there are 9 R-to-Q and 22 other R mutations in our dataset, and we see no significant difference between them in the distribution of effects on affinity or specificity.

All-in-all I have no problem recommending this paper for publication.

We thank the reviewer for their positive assessment of our manuscript, their noting the possibility of our analysis being used in the future for development of algorithms to improve specificity-change predictions, and their recommendation for publication in *Nature Communications*.

Reviewer #2 (Remarks to the Author):

Summary: In this paper, Kock et al use protein binding microarrays to assess how a large number of variants in human homeodomain (HD) proteins impact DNA binding affinity and specificity. To analyze the data, the authors developed new approaches to assess and visualize changes in binding to individual 8-mer sequences. Overall, the experiments and analysis are well performed, and the data support the idea that numerous different residues outside the classically defined DNA binding contact amino acid positions can influence both DNA binding affinity and/or specificity. These data serve as a valuable resource for those studying HD proteins. In addition, the data reveals that our current computational methods to predict how specific variants impact DNA specificity are insufficient to accurately assess changes in DNA binding and that additional empirical experiments coupled with predictive tools are needed to bridge this knowledge gap. For example, while the authors do identify new residue positions in the 60 amino acid HD that can impact DNA binding, they further found that other variants in these same positions can result in either no change in binding, an affinity change, a specificity change, or a novel specificity change. Hence, generalized rules for how variants in key amino acid positions impact DNA binding are not established in this study. Nevertheless, revealing this knowledge gap is valuable to the field and the authors provide an experimental platform and new way to analyze PBM data to highlight changes in DNA binding to 8mer sequences.

We thank the reviewer for their positive assessment of our manuscript, appreciating our data as a valuable resource for the community and noting that our study reveals an important knowledge gap in the field.

Below are a list of comments that should be addressed to both increase the ease of understanding the data and to better validate the results:

1) The authors list all tested factors by name and variant within the context of the full-length protein such as ALX3 L168V. However, it would also be helpful to include a reference to where the amino acid variant is located within the 60 amino acid HD such as ALX3 L168V (HD#XX) in both the Figures and the text. This would make it much easier to determine where the variant is within the HD and if it impacts the N-terminal ARM, Helix1, 2, or 3, etc without referring to Supplementary table 2.

We have implemented this excellent suggestion.

2) The guide graph in Figure 3A is very helpful. However, in Figure 4, the X-axis changes from contrastAverage to reference affinity. Are these the same? If so, consistent nomenclature should be used. If not, please consider creating a guide like in Figure 3A within Figure 4 to highlight what the "reference affinity" axis is and how it differs from contrast Average.

The difference in axes existed for historical reasons and we apologize for the confusion that it caused. We have resolved the issue by converting all plots to use the same axes; guide graphs are now provided in a new Figure 2 for both contrast and specificity plots.

3) Since the analysis of the PBM data is novel (upbm), it would be helpful to provide a more thorough explanation of the analysis. In addition to point #2 above, the following are suggestions:

- Supplementary figure 2 summarizes the key features of upbm. Consider incorporating this diagram or an abbreviated version of it into Figure 2.
- Explain whether the numbers along the X-axis of each graph are relative and therefore apply to just that control/variant or absolute and can be compared across different graphs/factors/variants. In other words, what is upbm affinity? Is it an absolute measurement or used primarily for relative comparisons?
- I became a little confused as to how to visualize the graphs and determine if a variant is selectively impacting the affinity to a subset of 8mers or generally impacting the affinity to all sites. Hence, it would be helpful if the authors showed clear example graphs of representative variants that : i) strongly disrupt all DNA binding compared to WT; ii) Weakly disrupt DNA binding affinity to all 8mers; versus iii) Disrupt DNA binding affinity to only a subset of 8mers. These examples would help the readers interpret the graphs in a more intuitive manner.

These suggestions have been incorporated in the new Figure 2. We are grateful for the reviewer's careful assistance in making the method more understandable. We have also addressed point b about upbm affinity in the main text.

4) Of the datasets that were reprocessed with the new upbm pipeline, it would be informative to include which alleles yielded different results from previous analysis/studies.

This information has been added to Supplementary Table 2 and briefly summarized in the main text.

5) In ChIP analyses, I am assuming that FG stands for foreground and BG stands for background, but I do not see a place where this is defined. Also, were only the top 1000 peaks used for analysis? If so, this needs to be mentioned in the main text. Lastly, it is unclear why the authors haven't used common, validated, and accepted methods in the field for this such as HOMER, MEME, or FIMO. I understand the authors wanted to search for only one sequence instead of a PWM but as far as I'm aware this is possible with the forementioned methods. These tools would apply to both human and mouse genomes whereas the mentioned method GENRE does not currently apply to mouse genomes. Further, why were different peak windows used in the CRX (200 bp) and SIX (80bp) ChIP-seq analyses?

We have reanalyzed the ChIP data using a newer version of GENRE, which permits uniform application across different genomes and with a common peak size (150 bp). We have added detail to the main text discussion of this analysis. We prefer GENRE over other methods because our lab has previously shown that a feature-matched background of genuine genomic sequence provides more sensitive detection of motif enrichment with reduced false positives compared to the background models implemented by other methods, including those suggested by the reviewer (Mariani *et al.*, *Cell Systems* 5:187, 2017).

6) The ability of upbm to detect a gain of specificity and/or affinity should be validated with another DNA binding method.

We have curated a collection of published data demonstrating the effects (or non-effects) on sequence-specific binding by HD variants included in this study; a total of 24 variant x sequence x publication combinations were available. The results of comparing this set to upbm output are summarized in the main text and presented in detail in a new Supplementary Table 4. Moreover, additional validation by reanalysis of previously published PBM data and comparison to orthogonal methods was inadvertently omitted from the current Supplementary Figure 2 and is now included.

7) Better explanation of the possible association of VUS alleles and disease is warranted. For example, while some variants tested in the paper were previously shown to be pathogenic, others were VUSs. In some sections of the paper, the authors make associations between their data and VUSs that are not well justified or need to be better justified based on available genetic data. A prime example is the finding that the PBX4 rare variant R215Q results in a severe loss of affinity. The authors subsequently argue that this variant may underly hematological and metabolic defects. But it is unclear if this variant was found in a patient or patients with such defects, and whether the patient(s) are heterozygous or homozygous for such variants. The authors should better integrate available genomic data that provides insight into whether VUS alleles of factors such as PBX4 are likely haploinsufficient or not. In scanning the gnomad database, PBX4 has a PLI of 0 – suggesting PBX4 is not haploinsufficient. Hence any argument that the R215Q allele is associated with disease would either require evidence of homozygous alleles in humans or a possible explanation of how the R215Q functions as a potential dominant negative allele. Hence, the human allele data, patient symptom data, and the PBM binding data using different alleles needs to be better explained and clarified.

To address this and related reviewer concerns, we have removed statements about disease relevance from our report of the results of individual *in vitro* experiments (while in some cases reporting evidence that a particular motif affected *in vitro* is also enriched in ChIP peaks, which we believe supports the idea that altered binding to that motif is likely to be functionally relevant *in vivo*), and instead added a statement to the Discussion highlighting the importance of functional data for prioritizing genetic studies of variants when the community is identifying them in unwieldy numbers.

8) The difference between yes/- and yes/+ in Supplementary Table 2 is unclear.

We have replaced these with “reduced” and “increased”, respectively.

9) What does replicate refer to in Supplementary Table 3? Is it the number of total replicates performed on each variant? If so, what is the justification between the different replicate numbers?

In most cases, experiments were replicated when we thought they looked questionable, based on a number of QC heuristics our lab has used over the years. After all the data were available, a number of standardized QC metrics, developed as part of the upbm package, were uniformly applied, as explained in the Methods, and any dataset that passed these filters was included.

10) Unclear where the following data is shown: “Our homology modeling results suggest that SIX6 Thr-165 at position 38 contacts Thr-139 at position 12, Leu-143 at position 16, and Leu-167 at position 40; intriguingly, these residues are within, or adjacent to, the hydrophobic core, and thus are not ones that would have been expected a priori to contribute to DNA binding specificity.”

A reference to Supplementary Figure 5 has been added to the text and additional panels have been added to that Supplementary Figure (panels i, j) to show the modeled interaction networks.

11) It would benefit the reader to show images of the crystal structure of some of the key residues (positions 58, 44, 25, 26, and 31) the authors are highlighting starting on page 14.

These residues are highlighted in Figure 6b.

12) While there are many panels in Supplementary Figure 4, adding labels such as A, B, C for each factor would make it much easier to find the referenced data throughout the paper without having to look through all the pages of graphs.

Panel labels have been added. Where the figure (now Supplementary Figure 3) is referenced as a whole, it is because there are several examples of a given phenomenon and this figure contains a graphical representation of the entire dataset.

Minor comments:

- Second half of the first paragraph of the introduction is the same statement word for word as the beginning of the abstract.
- It would be helpful for the reader to label every fifth residue (or more) in Figure 1b.
- Please list the four affinity-altering variants on page 10.
- Top of page 16, there should be a comma between 12 and ALX4 Q225E

We thank the reviewer for catching these points and have addressed all of these issues in the revised manuscript text.

Reviewer #3 (Remarks to the Author):

The manuscript entitled “DNA binding analysis of rare variants in homeodomains reveals novel homeodomain specificity-determining residues” presents a survey of DNA binding affinity and specificity profiles for 92 variants of homeodomain transcription factors. This study utilizes protein-binding microarrays (PBMs) which are analyzed using a modified pipeline (“upbm”) in order to assess binding to both high- and lower-affinity 8-mers and to better observe differences in specificity. As a large TF family with many disease associations, homeodomains are of significant academic and clinical relevance. Furthermore, variants tested include pathogenic and likely pathogenic variants associated with a variety of diseases. The authors identify 14 novel residues within homeodomain regions that affect DNA binding affinity or specificity. Of these 14 residues, five are not thought to contact DNA directly, a key result that supports recent findings that changes outside of DNA-contacting residues or even outside the borders of DNA binding domains can affect DNA binding of TFs. For many known and novel specificity-determining residues, the authors observe that lower-affinity sites, rather than the highest affinity sites, are often differentially bound by homeodomain variants. Importantly, this finding provides evidence for the emerging understanding that DNA occupancy of TFs is largely determined by the composite effect of several lower-affinity interactions. The authors highlight that DNA binding prediction tools are especially poor for predicting differences in specificity, illustrating the need for experimental studies such as this one. To this point, the authors observe that seemingly analogous mutations in different transcription factors can have distinct effects on DNA binding. Importantly, the authors support their findings using structural models to generate hypotheses about physiochemical interactions that may be involved in differential DNA binding.

Overall, this manuscript is very clear and well-written, facilitating easy understanding of the results. The findings are of significant interest to the transcription factor field, as they illustrate properties of TF-DNA binding that are of emerging importance and cannot currently be predicted using in silico approaches. The authors’ findings that mutations in homeodomain regions can have such distinct effects across different transcription factors suggest that similar or more marked differences may also exist within TF families with more variable DNA binding domains. Considering the clinical relevance of the mutations studied, the manuscript also provides valuable insight into mechanisms connecting rare variants and their associated clinical phenotypes. Due to all these reasons, I recommend publication of the article

following some minor revisions suggested below to improve clarity and the precision of stated conclusions.

We thank the reviewer for their positive assessment of our manuscript, noting that our findings are of significant interest to the transcription factor field and that our manuscript provides valuable insight into mechanisms connecting rare variants and their associated clinical phenotypes, and recommending our manuscript for publication after some minor, suggested revisions.

1) It would be useful to add the frequency of the variants tested in Table S2.

This information has now been added, for those variants for which it is available.

2) Introduction, paragraph 4: “upm” should be “upbm.”

We thank the reviewer for catching this typo. It has been fixed in the revised manuscript text.

3) In Fig. 1 it is somewhat difficult to visualize which residues correspond to which bars as the 3 panels are difficult to align. Consider using a heatmap with a layout similar to Fig. 5a.

A heatmap requires the distinction of shades of color and is thus more difficult for readers with compromised color vision. We have added additional position labels and extended gridlines to improve readability of the aligned display items.

4) More information should be reported in the Results section and maybe represented in a main figure regarding identity of the 92 variants tested, namely the number of distinct TFs represented per TF and subfamily and the number of variants of each type (pathogenic, likely pathogenic, VUS). Although this information is present in Table S2, it may be convenient for the reader to have it available in the text and figure.

Subfamily distribution of the tested variants has been added to Supplementary Figure 1.

5) Supp. Fig. 3: This figure seems to be misnumbered and missing panels referenced in the text. Panels referring to HOXD3 are listed as Supp. Fig. 3j-m in the text but are shown in panels b-e in the figure. Panels n,o in the text should be f and g (these panels are also hard to read in the figure). Relatedly, The text mentions a comparison between Msn2 and Msn4 in Supplementary Fig. 3 b-e but these are not included in the figure; additionally, Msn4 is not mentioned earlier in the text along with Msn2, Com2, and Usv1 as a TF that was studied, so it's unclear where Msn4 comes into play.

This (now Supplementary Figure 2) was an unfortunate oversight, the uploading of an incorrect draft version, for which we apologize. The complete version is now included. The failure to mention Msn4 and also Rgm1 in the main text has also been corrected.

6) Fig. 2a: Consider replacing this pie chart with something more intuitive to help the reader more easily interpret the result. Since two variables are being considered (affinity and specificity), it may help to represent these variables separately (e.g., as a contingency table, heatmap, or similar). This is just a recommendation and it's up to the authors whether changes are made.

We tried these alternate formats but find this one more readable.

7) Fig. 2b-d: Since the less successful prediction tools are discussed in the main text, the reader may

expect these to also be represented in the main figure panels. While the figure caption explains that only the top 10 most successful tools are shown, it may help to explain this in the text as well to avoid confusion. Alternatively, consider the benefits of including results for all tools (as in Supp. Fig. 5) in the main figure, perhaps with the top 10 best tools with thicker lines and the remaining ones with thin lines. This is just a recommendation and it's up to the authors whether changes are made.

All tools are now included in main body ROC plots, with additional information (PRC plots and AUC information for both) included in the Supplementary Figure (now Supplementary Figure 4).

8) Fig. 3: Throughout the data panels b-g displaying contrastDifference and contrastAverage, it is somewhat easy to identify decreased affinity, but changes in specificity are harder to observe for some examples mentioned. For example, it is difficult to see why ALX3 L168V reduces affinity while HOXA4 R217L reduces affinity and also changes specificity when the plots look somewhat similar. It might help to highlight the regions in the real case plots that show difference in specificity.

All data points in this figure are colored by contrast, not specificity, to highlight affinity differences. We have added a reference to the appropriate panel of Supplementary Figure 3 (which shows both plots for all variants reported) to make this visible without drastically altering the flow of the text.

9) P10 top paragraph: "Finally, 16 of 32 missense variants that were present only in gnomAD..." It would be helpful to be more explicit about how you are interpreting a variant that is only present in gnomAD (i.e., that are common variants and/or have no clinical association, or if rare they may affect function).

Please see the response to Reviewer 2's point 7.

10) P10 bottom paragraph: This paragraph talks extensively about structural details, including mentions of a hydrophobic core and a helix 1 proline substitution. It may help to use structural figures/models here to introduce and help the reader to visualize these important structural features.

A new Figure 1a introduces the homeodomain structure and highlights the features that we discuss later in the manuscript.

11) Fig. 3g: "partial sparing of the highest-affinity 8-mers" by HESX1 R109Q is not apparent to the inexperienced reader from looking at the figure. Consider some indications in the figure to help illustrate this point.

New labels have been added to address this and the subsequent point.

12) Fig. 4a: Consider also highlighting "retained high-affinity 8-mers" that are mentioned in the results text, as they are not apparent in the figure.

See above response.

13) P11: "suggesting... that this variant may also lead to cranial structural birth defects" seems like an overstatement as this study does not identify molecular or phenotypic outcomes of this variant. Similarly, with the NKX2-6 R150C variant in P10. Consider toning this down, or moving the discussion of potential disease relevance to the Discussion section.

Please see the response to Reviewer 2's point 7.

14) Fig. 4c,g,l: Enrichment of motifs within ChIP-seq peaks is not easily interpreted from these graphs. Consider alternative representations, e.g. by representing proportions. Also, define “FG” and “BG” in the figure and caption.

We considered instead presenting contingency tables, but feel that the visual depiction is clearer. We have attempted to improve the labeling. Please note this figure has been moved to Figure 5 in the revised manuscript.

15) Fig. 4e-g: It is not clear if or how ChIP-seq binding of wildtype SIX6 to the noncanonical TCACAC site suggests that “SIX6 T165A may exhibit enhanced binding at these sites in vivo, and that gain of specificity by SIX6 variants may be pathogenic.” Consider alternative and toned-down interpretations. Additionally, it is unclear why the T165G mutation was also tested and what conclusions can be drawn from it.

Please see the response to Reviewer 2’s point 7.

16) In Fig. 3, PBM results are represented with “contrastAverage” as the X-axis variable. However, Figs. 4 and 5 use “reference affinity” as the X-axis variable. It’s not clear why this variable is not consistent across all figures or if this changes how readers should interpret the panels in Fig. 3 vs. Figs. 4 and 5.

As discussed in response to Reviewer 2’s comments, this was an unfortunate historical holdover and has now been revised, with all upbm plots using the same axes in the resubmission.

17) Fig. 4h,i: The results text states that VENTX R121Q and Y115F show preferential loss of binding to lower-affinity sites, but this is not clear to the average reader in the figure panels, where the highlighted 8-mers appear to correspond to higher-affinity sites on the far right side of the graph. If possible, this should somehow be made more clear to the reader.

New labels have been added to the figure panels to address this point. Please note this figure has been moved to Figure 5 in the revised manuscript.

18) P13, top paragraph: “...suggesting that HD mutants that perturb binding to lower affinity sites may affect gene regulation” may be an overstatement as the data mentioned only pertain to TF-DNA binding and not to recruitment of cofactors/transcriptional machinery or changes in transcription.

Please see the response to Reviewer 2’s point 7.

19) P14 bottom paragraph: “...suggesting indirect readout of the underlying DNA sequence”; it is not clear what is meant by “indirect readout” or how the PBM results suggest this.

20) P20, top paragraph: Here the authors mention that a main goal was “...to explore potential context-specific effects” of a mutation in the same residue in different TFs. Consider commenting on/reminding the reader about what context features may affect the outcome of a mutation.

We have expanded on both of the above two points in the main text and tried to clarify our line of thinking. “Indirect readout” was defined in a landmark *Nature* (1988) paper [PMID: 3419502]; we have provided additional clarifying text and cite that paper in the revised manuscript.

21) Fig. 5a “current study variants”: Consider as a heat map to show the proportion of variants at each position that display each type of change (alternatively, add the number in each cell). Also, consider using a different color (grey) to indicate positions where no variant was tested, so the reader can differentiate between residues not tested and residues that did not appear to have an effect.

We consider the figure rather dense already and have not added an additional row. Any position that is white in all 4 result rows (No change, No affinity change, No specificity change, Not a novel specificity change) was necessarily untested. The number of variants tested and proportion showing affinity and specificity change, by position, has been added as an additional tab in Supplementary Table 2.

Juan Fuxman Bass

Thank you, Juan, for your time and effort on this review!

REVIEWERS' COMMENTS

Reviewer #1 (Remarks to the Author):

I appreciate author's consideration of my feedback. I think one sentence-initial numeral was overlooked: line 242 p11: "47 of these are either...". And two others occur after semicolons that would be better replaced with periods: line 337 p15: "...substitution; 2 of the other 3..." and line 354 p16: "...substitution; 2 of the other 3...". Both semicolons are unnecessary, as they make the total sentence length (before + after) a whopping 7 lines. After replacing those semicolons with periods, the initial numerals 2 should be replaced with Two.

Implementation of reviewer 2's excellent suggestion to include the core HD position for all mutations makes it easier for the reader to determine which mutations are more or less equivalent. Much appreciated.

After addressing the minor points above, I can recommend publication of this manuscript.

Reviewer #2 (Remarks to the Author):

The authors did a great job in revising the manuscript. All the main/major criticisms have been addressed. It is a very fun paper to read and the quality of the data and analysis are outstanding. I highly recommend acceptance for publication.

Reviewer #3 (Remarks to the Author):

The revised manuscript has addressed my comments and therefore, I recommend for publication in Nature Communications.

Reviewer #1 (Remarks to the Author):

I appreciate author's consideration of my feedback. I think one sentence-initial numeral was overlooked: line 242 p11: "47 of these are either...". And two others occur after semicolons that would be better replaced with periods: line 337 p15: "...substitution; 2 of the other 3..." and line 354 p16: "...substitution; 2 of the other 3...". Both semicolons are unnecessary, as they make the total sentence length (before + after) a whopping 7 lines. After replacing those semicolons with periods, the initial numerals 2 should be replaced with Two. Implementation of reviewer 2's excellent suggestion to include the core HD position for all mutations makes it easier for the reader to determine which mutations are more or less equivalent. Much appreciated.

After addressing the minor points above, I can recommend publication of this manuscript.

We apologize for missing the sentence-initial numeral on p. 11 and have corrected it. We have removed additional semicolons; each one remaining connects two independent clauses that collectively express a single idea.

Reviewer #2 (Remarks to the Author):

The authors did a great job in revising the manuscript. All the main/major criticisms have been addressed. It is a very fun paper to read and the quality of the data and analysis are outstanding. I highly recommend acceptance for publication.

We thank the reviewer for the questions that we feel made for a better paper.

Reviewer #3 (Remarks to the Author):

The revised manuscript has addressed my comments and therefore, I recommend for publication in Nature Communications.

We thank the reviewer for the extremely helpful feedback.